# September Arctic Sea Ice minimum prediction – a new skillful statistical approach

Monica Ionita[1,2], Klaus Grosfeld[1], Patrick Scholz[1], Renate Treffeisen[1], Gerrit Lohmann[1,2]

[1] Alfred Wegener Institute Helmholtz Center for Polar and Marine Research, Bremerhaven, 27570, Germany
[2] MARUM – Center for Marine Environmental Sciences, University of Bremen, Bremen, Germany

*Correspondence to*: Monica Ionita (Monica.Ionita@awi.de)

**Abstract.** Sea ice in both Polar Regions is an important indicator for the expression of global climate change and its polar amplification. Consequently, a broad interest exists on sea ice coverage, variability and long term change. However, its predictability is complex and it depends strongly on different atmospheric and oceanic parameters. In order to provide insights into the potential development of a monthly/seasonal signal of sea ice evolution, we applied a robust statistical model based on different oceanic and atmospheric parameters to calculate an estimate of the September sea ice extent (SSIE) on monthly time scale. Although previous statistical attempts of monthly/seasonal SSIE forecasts show a relatively reduced skill, when the trend is removed, we show here that the September sea ice extent has a high predictive skill, up to 4 months ahead, based on previous months' oceanic and atmospheric conditions. Our statistical model skillfully captures the interannual variability of the SSIE and could provide a valuable tool for identifying relevant regions and oceanic and atmospheric parameters that are important for the sea ice development in the Arctic and for detecting sensitive/critical regions in global coupled climate models with focus on sea ice formation.

## 1 Introduction

Arctic sea ice plays an important role in modulating the global climate system by influencing the atmospheric and oceanic circulation in Polar Regions. Moreover, it has a strong impact also on the global economic system through changes in marine and natural resources development. The sea ice extent over the Arctic region has undergone an extraordinary decline during the last decades that can be linked to climate change (Allison et al., 2009; Kay et al., 2011; Notz and Marotzke, 2012, Stroeve and Notz, 2018). The trends in the Arctic sea-ice extent are negative for all months, with the largest trend recorded at the end of the melt season in September (Serreze et al., 2007), with an average decline of 12.9% per decade relative to the long-term mean of 1981-2010 September average (Cavalieri and Parkinson, 2012; Comiso et al., 2017). These negative trends, with their environmental and economic implications as well as its impacts on human society, have led to a rising demand for accurate sea ice predictions at monthly, seasonal up to decadal time scales, which in turn will be able to address the growing demands from different stakeholders and the scientific community (Meier et al., 2014). As such, an accurate sea ice prediction plays a crucial role for ecosystems, coastal communities, planning for new shipping ports, oil and gas exploration and marine transportation. The ten lowest September sea ice extents all occurred in the past 10 years and climate projections indicate that the Arctic Ocean could be ice free (sea ice less than $1x10^6$ km² for at least five consecutive years) in September in the second half of the 21[st] century (IPCC, 2013) . As a result, the ship traffic and Arctic resources extraction have already increased (Pizzolato et al., 2014). For example, the exploitation of shipping via the Northwest Passage or Northeast Passage could reduce the

navigational distance between Europe and Asia by ~40% compared to the route via the Suez canal (Schøyen and Bråthen, 2011). The reduction in distance compared to the Suez and/or Panama Canal routes could result in large cost saving due to reduced fuel consumption and an increase in the number of ships (Lassere, 2015). Melia et al. (2016) have shown that by mid-century, the frequency of navigable period will double and the routes across the central Arctic will become available. For example, for a high-emission scenario they have shown that by late century trans-Arctic shipping might become a commonplace, with the shipping season ranging from four to eight months. Overall, the summertime use of these routes by different vessels (i.e. cargo ship and tanks) has increased (Eguíluz et al., 2016), thus the need for a proper forecast for the Arctic sea ice conditions has become imperative. Currently, forecasting the open water route through the Arctic basin is accurate within 200 km when the predictions are initialized in July (Melia et al., 2016). As such, an early knowledge on the potential opening of the maritime Arctic routes could allow a better management for the shipping companies to optimize (in terms of time and costs) shipping routes between the Atlantic and the Pacific Oceans (Hassol, 2004; Smith and Stephenson, 2013). However, the opening of the Northeast and Northwest Passages does not guarantee ice free transects along the passages at all times and can always include the possibility of drifting ice floes, which for conventional ships poses high risks and potential environmental danger when getting damaged in case of accidents. Pizzolato et al. (2016) have shown, that despite the persistence of low sea ice condition since 2007, very little shipping activities has been recorded within the northern route of the Northwest Passage. This might be attributed to the multiyear ice concentrations in the Canadian Arctic waters, which strongly influences the shipping activity. Hence, a proper forecast does not imply a dangerous free transect as long as the Arctic Ocean is ice covered with thick multiyear ice for its larger parts over the significant times of the year.

Although the evolution of Arctic sea ice physical properties has been extensively studied, the prediction of detrended Arctic sea ice extent, with lead times of 3 months and longer, have not been very promising (Lindsay et al., 2008; Blanchard- Wrigglesworth et al., 2011). From a forecasting point of view, the evolution of autumn Arctic sea ice is closely associated with initial conditions in previous winter and spring. Different studies have emphasized that some parameters contribute significantly to the improvement of the seasonal sea ice forecast skill at different time lags (Holland and Stroeve, 2011; Lindsay et al., 2008). For example, sea surface temperature and sea ice concentration in spring are highly relevant predictors for the minimum Arctic sea ice extent (Drobot et al., 2006). Some studies suggested that accurate sea ice thickness could increase the forecast skill 2 months ahead (Day et al., 2014; Dirkson et al., 2017). Also, the spring melt pond fraction has been employed to improve the forecast skill of the Arctic minimum sea ice extent (Schröder et al., 2014).

Currently, there are different approaches used to make sea ice forecast: ice-ocean-atmosphere coupled models, statistical models, best guesses model and mixed models (Stroeve et al., 2014; Hamilton and Stroeve, 2016). From a statistical point of view, Drobot et al. (2006) showed that 46% of the pan-Arctic minimum sea ice extent would be predictable as early as February based on monthly sea ice concentration, surface albedo, downwelling long-wave radiation and surface skin temperature. Lindsay et al. (2008) have shown that their statistical model based on a wide range of predictors (e.g., atmospheric circulation indices, sea ice extent and sea ice concentration, ocean temperature at different levels) exhibited a greater skill in predicting the September sea ice extent (SSIE) than those by Drobot et al. (2006). The forecasts based on the state-of-the-art coupled atmosphere-ocean sea ice models (Chevallier et al., 2013; Sigmond et al., 2013) do not show better results when compared

with the statistical models (Kapsch et al., 2014; Schröder et al., 2014; Zhan and Davies, 2017). These caveats indicate that our understanding regarding the controlling factors of Arctic sea ice may still be insufficient. Overall, skillful forecasts extend only two to five months ahead, for the summer months (Stroeve et al., 2015; Schröder et al., 2014), regardless of the type of the model used for the forecast (dynamical or statistical). The results and error margins based on these different approaches have highlighted how difficult it is to make skillful prediction for the SSIE. This is particular true for the years with extreme low September sea ice concentrations (e.g., 2012 or 2007), with both, the dynamical and the statistical models showing similar limitations (Stroeve et al., 2015; Schröder et al., 2014; Stroeve et al., 2014; Hamilton and Stroeve, 2016). Stroeve et al. (2014) have shown that seasonal predictions of the SSIE are most accurate in years when the sea ice extent is near the long-term trend, but skillful sea ice extent prediction appear challenging in years when the weather plays a larger role (Hamilton and Stroeve, 2016).

In order to improve the monthly/seasonal prediction skill of the sea ice extent one possibility would be to identify stable predictors (the correlation coefficient between the predictor and the predictand does not change in time) and to develop a statistical forecast model based on these predictors. Following this idea, here we analyze the oceanic and atmospheric conditions associated to the SSIE in order to identify potential predictors based on a simple statistical methodology and placed them in a longer temporal context. Our statistical model takes into account different atmospheric and oceanic variables following the approach in Ionita et al. (2008, 2014, 2017, 2018). These parameters are: sea level pressure (SLP), air temperature (TT), precipitable water content (PWC), surface zonal wind (USURF), surface meridional wind (VSURF), the ocean heat content integrated over the first 700m (OHC), sea surface temperature (SST) and water temperature integrated over the first 100m (OT100), in order to calculate an estimate of SSIE. The paper is structured as follows: the data and methods used in this study are presented in Section 2, while the main results of our analysis are shown in Section 3. The discussion and concluding remarks are presented in Section 4 and 5.

## 2 Data and methods

### 2.1 Data

The monthly sea ice extent has been extracted from the National Snow and Ice Data Center ftp server (ftp://sidads.colorado.edu/DATASETS/NOAA/G02135/north/) (Fetterer et al., 2016).

For the Northern Hemisphere temperature and atmospheric circulation, we use the monthly means of air temperature at 2m (TT), downward longwave radiation flux (DW), zonal wind (USURF), meridional wind (VSURF), precipitable water content (PWC) and the mean sea level pressure (SLP) from the NCEP/NCAR 40-year reanalysis project (Kalnay et al., 1996) on a 2.5º x 2.5º grid. Global sea surface temperature (SST) is extracted from the Extended Reconstructed Sea Surface Temperature data (ERSSTv5) (Huang et al., 2014). This dataset covers the period 1854 – present and has a spatial resolution of 2° x 2°. The global heat content data in the first 700m (OHC) and the ocean temperature integrated over the first 100m (OT100) is extracted from the Global Ocean Heat and Salt Content database (Levitus et al., 2012; Boyer et al., 2013).

The monthly Atlantic Multidecadal Oscillation (AMO) index The AMO index has been calculated as the average of monthly SST anomalies with respect to the mean over the North Atlantic north of 25◦N (75◦W–7◦W, 25◦N–60◦N). For the AMO index computation, we used the RRSSTv5 data set (Huang et al., 2014). In this study, we

use the yearly mean of AMO index. Table 1 gives an overview of all the data sets included in the study. All used data sets have been detrended before the analysis by computing the linear trend for the entire time series/gridded fields in question. This trend was then subtracted from the initial time series/gridded data set. The linear trend was estimated using a least-square linear regression.

## 2.2 Stability Maps

The statistical model used in this study for the estimation of SSIE is based on a methodology successfully used to make monthly/seasonal streamflow predictions for the central European rivers (e.g., Elbe river, Rhine river, Danube river, Ionita et al., 2008, 2014, 2017, 2018; Meißner et al., 2017). Furthermore, for identifying the drivers

of the Antarctic sea ice variability (Ionita et al., 2018). The basic idea of this method is to identify regions where the spatio-temporal distribution of the predictors is stable correlated with the Pan-Arctic SSIE. The SSIE has been correlated with the potential predictors from previous months (Table 2) in a moving window of 21 years and the statistical significance of the correlation coefficient was tested using a two-sided *Student t-test*. The correlation is considered *stable* for those grid-points where SSIE and the large-scale predictors (e.g., OHC,

OT100, SST, SLP, TT, PWC, DW, USURF and VSURF) are significantly correlated at 95%, 90%, 85% and 80% significance level for more than 80% of the 21-year windows, covering the period 1979-2007. We choose the period 1979-2007 as calibration period, as both extreme years of sea ice extent, namely 1996 and 2007, were included and it provides a climate relevant period of nearly 30 years. The area where the correlation coefficient is stable and positive are represented as dark red (95%), red (90%), orange (85%) and yellow (80%), while the

regions where correlation coefficient is stable and negative are represented as dark blue (95%), blue (90%), green (85%) and light green (80%). Such maps are referred in our study as *stability maps* and their spatial structures remain qualitatively the same if the significance levels that define the stability of the correlation vary within reasonable limits and if the length of the moving window varies between 15 to 25 years. The optimal predictors are defined as the average values over the stable regions for each gridded parameter. For the current analysis only

regions where the correlation is above 90% significance level, are retained for further analysis (Figure 1). The raw stability maps between SSIE (pan-Arctic and regional) and the potential predictors are shown in Figures S3 – S15. Although the length of our time series is relatively short (40 years) the methodology proved to work also in cases of timer series <40 years (Ionita et al., 2018). Moreover, we use the same methodology, with the same number of years (40 years), for the prediction of September Arctic and Antarctic Sea Ice

(https://www.arcus.org/sipn/sea-ice-outlook/2017/post-season).

As a further main contributor to our forecast model, we use persistence, defined here as the sea ice extent from previous months (e.g., January, February up to August). Persistence of sea ice anomalies stands as the first source of predictability for sea ice (Guemas et al., 2016; Walsh et al., 1979; Blanchard-Wrigglesworth et al., 2011).

## 2.3 Multiple Linear Regression

For the forecast all datasets were separated into two parts: 1) the calibration period (1979–2007) and 2) the validation period (2008 – 2017). The optimal predictors are identified by employing stepwise multiple regression analysis (e.g., von Storch and Zwiers, 1999). Although the "*stability maps*" methodology (Figure 1) identifies multiple stable regions for each atmospheric/oceanic parameter (Figures S3 – S15), after applying the stepwise

multiple regression, the optimal/final prediction model is based just on the regions shown in Figures 1 - 3 and 5 - 7. To forecast the September Sea Ice Extent we have used a multiple linear regression model with the regression equation:

$$Y = \beta_0 + \beta_1 x_1 + \beta_2 x_2 + \cdots + \beta_n x_n + \varepsilon$$

where $Y$ represents the SSIE, $\beta_0$, $\beta_1$, $\beta_2$,...$\beta_n$ are constants determined by the least squares procedure, $x_1$, $x_2$,...$x_n$ the predictors used (e.g., OHC, OT100, etc) and $\varepsilon$ the error.

In this study, we choose stepwise regression. Thus, each predictor was prioritized based on its correlation coefficient with the SSIE and was added to the model in that order. As we added more predictors to the model, the $F$ statistic was used to determine whether the added predictors were significant in the regression equation. Entrance and exit criteria for the $F$ statistic were set to 0.05 and 0.1, respectively. Stepwise regression was used because it prioritizes predictors based on the partial correlation and it is likely that high and significant correlations will reflect underlying physical processes. In order to estimate possible over fitting, we make use of the Akaike information criterion (AIC) (von Storch and Zwiers, 1999), the explained variance, $R^2$ and the residual standard error. A workflow of the selection of the optimal model for the SSIE prediction is shown in the supplementary file and Figure S2.

## 3 Results

### 3.1 Pan-Arctic September sea ice prediction

The skill of a long-range forecast for the Arctic SSIE is associated with the predictors that represent the slow varying components of the climate system that are able to integrate the climate information such as ocean heat content and SST (Guemas et al., 2016, Lindsay et al., 2008). These variables can be used as potential predictors for months and even seasons in advance due to their long-term memory. Thus, here we investigate the potential link between the Arctic SSIE (Fetterer et al., 2016) and OHC, OT100 (Levitus et al., 2012; Boyer et al., 2013) and SST (Huang et al., 2014) as long-term predictors (lags ~4 years (AMO index) up to 2 months in advance, see Table 2 for a detailed description of all the lags used in the study). On shorter time-scales (2 – 4 months) the atmospheric circulation, especially during the summer months, plays a major role in driving the Arctic sea ice variability (Guemas et al., 2016). The atmospheric circulation can substantially contribute to the skill of the sea ice predictions. As such, for the SSIE prediction we have also tested the skill of atmospheric variables (up to 4 months in advance), e.g., SLP, TT, PWC, USURF and VSURF (Kalnay et al., 1996). Atmospheric moisture content (e.g., clouds, water vapor content) has an impact on the net surface radiation balance and hence also on the SSIE (Kapsch et al., 2013, 2014). As a measure for this impact, we use the precipitable water content (PWC) as an additional predictor.

For the final forecast, based on data available at the end of May (4 months ahead forecast) we have retained all identified stable regions shown as black boxes in Figure 1. For the forecast based on June data, we have included also the stable regions based on all June stability maps (Figure 2). We have applied the same technique for the July data (Figure 3). For SSIE prediction based on the end of May data, the optimal model is based on a combination of: OHC SON, SST MAM, PWC Apr, VSURF MAM and SLP May (Table 3). Together with these identified stable regions, the optimal model includes also the persistence of sea ice extent (here the sea ice extent from previous March (SIE Mar), as well as the annual Atlantic Multidecadal Oscillation index, with a lag 4 of

years (AMO L4). The highest correlation between SSIE and the annual AMO index was found at a time lag of 4 years (AMO leads SSIE). The time lag identified in our analysis is in line with previous studies (Day et al., 2012; Mahajan et al., 2011). The observed and forecasted values based on the May data are shown in Figure 4a. The explained variance of the model, over the calibration (*validation*) period, is 81% (*71%*) and the correlation coefficient between the observed and forecasted SSIE is r = 0.90 (*r = 84*) (99.9% significance level). To better assess the skill of the SSIE prediction, the root mean square error (RMSE), the Nush-Sutcliffe efficiency (NSE) and the index of agreement (d) are calculated, among other statistical tests (see Table S1 and supplementary file for a definition of all the metrics used to test the skill of the model). The forecasted model based on May data shows a very good skill (Table S1) NSE = 0.82 (*0.68*) (NSE = 1 means perfect model) and d = 0.95 (*0.88*) (d = 1 indicates a perfect match between the observed and forecasted values, d = 0 indicates no agreement at all).

Following the same steps as in the case of May data, for the model based on June data, the parameters contributing to the optimal forecast model are shown in Figure 2. As additional predictors, on top of those for May (Figure 1), we have: VSURF Jun, USURF Jun and TT Jun (Table 3). The observed and forecasted values of SSIE based on June data are shown in Figure 4b. The overall explained variance of the June-based model, over the calibration (*validation*) period, is 85% (*79%*) and the correlation coefficient between the observed and forecasted SSIE values is r = 0.92 (*r = 0.89*) (99.9% significance level). The June-based model exhibits also a very good skill and shows slight improvements compared to the May – based model (NSE = 0.85 (*0.78*) and d = 0.96 (*0.93*). For the model based on July data, the parameters contributing to the optimal forecast model, on top of those based on May (Figure 1) and June (Figure 2), are shown in Figure 3 and Table 3. The observed and predicted values of SSIE based on July data are shown in Figure 4c. The overall explained variance of the June-based model, over the calibration (*validation*) period, is 86% (*81%*) and the correlation coefficient between the observed and forecasted SSIE values is r = 0.93 (*r = 0.90*) (99.9% significance level). The July-based model exhibits also a very good skill and shows also slight improvements compared to the May and June – based models (NSE = 0.86 (*0.80*) and d = 0.96 (*0.94*)).

## 3.2 Application of the methodology for regional SSIE prediction

To test the robustness of our statistical model and to move towards stakeholder-relevant regions, in this study we are investigating also the skill of our model at regional scale. Thus, we have repeated the same analysis as in the previous section but for the sea ice extent averaged over the East Siberian Sea (ESS) (Figure S1). In this study, we focus on the ESS because in September 2007 and 2012, negative ice concentration anomalies were particularly pronounced over this region of the Arctic Ocean (Figure S1a and S1b, respectively) and the highest variability of the SSIE is recorded here (Figure S1c).In addition, since 2011 the eastern ESS has been nearly ice-free (<10% SSIE) at the end of summer (Polyakov et al., 2017). Moreover, when looking at the correlation coefficients between the pan-Arctic SSIE and regional September SIE, the highest correlation, at lag 0, is found with the ESS-SIE (r = 0.72, Table 4).

The stability maps between the detrended ESS SSIE and the large scale oceanic and atmospheric fields are shown in Figure 5 (stability maps based on May and previous months data), Figure 6 (stability maps based on June and previous months data) and Figure 7 (stability maps based on July and previous months data), respectively. For ESS SSIE prediction based on the end of May data, the optimal model is based on a combination of: annual

OT100, SST MAM, SLP Jan, VSURF MAM, PWC May, TT May and DW MAM (Table 5). The observed and forecasted values based on the May data are shown in Figure 8a. The explained variance of the model, over the calibration (*validation*) period, is 88% (*58%*) and the correlation coefficient between the observed and forecasted ESS SSIE is r = 0.94 (*r = 0.77*) (99.9% significance level). The forecasted model based on the May shows a very good skill (Table S2) NSE = 0.88 (*0.57*) (NSE = 1 means perfect model) and d = 0.97 (*0.86*) (d = 1 indicates a perfect match between the observed and forecasted values, d = 0 indicates no agreement at all).

For the model based on June data, the parameters contributing to the optimal forecast model in addition to the May variables are shown in Figure 6 and Table 5. As additional predictors, on top of May data (Figure 5), we have: SIE Jun, and TT Jun (Table 5). The observed and forecasted values of ESS-SSIE based on June data are shown in Figure 8b. The overall explained variance of the June-based model, over the calibration (*validation*) period, is 91% (*71%*) and the correlation coefficient between the observed and forecasted SSIE values is r = 0.95 (*r = 0.84*) (99.9% significance level). The June-based model exhibits also a very good skill and shows slight improvements compared to the May – based model (NSE = 0.91 (*0.69*) and d = 0.98 (*0.91*). For the model based on July data, the parameters contributing to the optimal forecast model, on top of May data (Figure 5) and June data (Figure 6), are shown in Figure 7 and Table 5. The observed and predicted values of SSIE based on July data are shown in Figure 8c. The overall explained variance of the July-based model, over the calibration (*validation*) period, is 94% (81%) and the correlation coefficient between the observed and forecasted SSIE values is r = 0.97 (*r = 0.90*) (99.9% significance level). The July-based model exhibits also a very good skill and shows slight improvements compared to the May and June – based models (NSE = 0.94 (*0.78*) and d = 0.98 (*0.93*)).

**4 Discussion**

The results of this study demonstrate that statistically based models are able to predict SSIE with high skill, if the accurate drivers and their regional localizations (herein stable regions) are identified via various statistical techniques. In this paper, our analysis was focused on a single month - September, but the same methodology has been be successfully applied also for other months/seasons and also for the Antarctic region (Ionita et al., 2018). Our results highlight the potential for skillful prediction of SSIE, both at pan-Arctic level as well as for ESS, based on large-scale drivers from stable regions. The ocean drivers (OHC, TT100 and SST) from the identified stable regions are strongly related with the Atlantic inflow or with the SST variability over regions strongly influenced by decadal modes of variability (e.g., Pacific Decadal Oscillation (PDO) in the central and north Pacific) to multidecadal modes of variability (e.g., Atlantic Multidecadal Oscillation (AMO) in the Atlantic Ocean region). The Atlantic inflow, AMO and PDO play a significant role in driving the Arctic sea ice variability (Polyakov et al., 2017; Miles et al., 2014; Ionita et al., 2016; Screen et al., 2016). For example, the North Atlantic might act as a source for the OHC anomaly identified over the Kara Sea, Laptev Sea and ESS (Figure 1 and Figure 5), thus contributing to the skill of our forecast. The OHC anomalies form the North Atlantic flow into the Arctic basin, via advection, affect the sea ice distribution (Polyakov et al., 2017, Ono et al., 2018). In a recent study, Yu et al., (2017) have shown that the leading mode of variability of global sea-ice concentration is positively correlated with the AMO and negatively correlated with the PDO. Furthermore, two thirds of the total global sea ice trend can be explained by a combination of these two modes of variability. Over-imposed on the interannual variability, the temperature and salinity of the Atlantic inflows to the Arctic Ocean shows also

pronounced decadal to multidecadal variability (Zhang, 2015). This aligns with the concept of different previous studies, which suggest that the decreasing trend in the Artic sea ice is partially driven by AMO (Park and Latif, 2008; Lindsay et al., 2005; Ding et al., 2014; Yu et al., 2017). Moreover, starting at the beginning of 1990's the AMO has switched to a positive phase, at the same time when the Arctic sea ice extent started its abrupt decline.

Thus, in this study we have tested previous years AMO index as a potential driver of the Arctic sea ice extent.

The stability maps based on the predictors related to the atmospheric variables (Figures 1-3) show significant and stable correlations with regions restricted to the Artic basin, indicating a much regional connection between the September sea ice variability and large-scale atmospheric circulation. The state of the Arctic SSIE depends both on the state of the ice in spring as well as on the atmospheric condition during summer (Ding et al., 2017). In this

respect, the precipitable water content and air temperature in spring and early summer were found to show significant predictive skill for the SSIE both at pan-Arctic as well as regional level. This is also in agreement with previous studies (Kapsch et al., 2013; 2014) which have shown a significantly increased cloudiness and humidity over the Arctic region in spring, thus accelerating the sea ice retreat in the upcoming summer, via enhanced longwave radiation.

Overall, such a methodology can be valuable also for the modelling community. If the coupled models, used for forecasting purposes, face problems to simulate the ocean and/or the climate background over the areas that play a significant role in driving the SSIE variability (stable regions), one expects a relatively small forecast skill. The opposite case is also valid: a good representation of the key regions that drive SSIE could imply a good forecast skill. For example, Parkinson et al. (2006) determined that many climate models tend to simulate more winter sea

ice in the Barents Sea compared to observations. One hypothesis for this overestimation is that the models underestimate the heat content in the Atlantic Basin (which has proved to be one of the main contributors for a skillful prediction for SSIE in our model). By using a simple and computationally inexpensive statistical approach, one can anticipate more than 80% of SSIE up to four months in advance, based on the antecedent atmospheric and oceanic conditions over stable regions. Moreover, our statistical model is able to properly

reproduce the years with extreme low / high sea ice extent, both at pan-Arctic level as well as at regional scale (e.g., 2007 and 2012 – low SSIE and 1996 – high SSIE; see Figure 4 and Figure 8). The predictability of these extreme years poses big challenges for the sea ice prediction community (Hamilton and Stroeve, 2016).

For example, one of the most unpredictable years was 2012. Most of the models (statistical and dynamical) were unable to properly forecast the extremely low value of the sea ice extent in September 2012 (Stroeve et al., 2014).

Overall, the statistical predictions came closer to the unexpected low sea ice extent in September 2012 than the dynamical-based predictions. In this respect, our statistical model was able to capture the overall decline in the SSIE and we forecasted the lowest sea ice extent since the observational period (Figure 4). Nevertheless, in terms of amplitude, our forecast has underestimated the observed values (Figure 4). One of the reasons for this underestimation could come from the fact that in August 2012 a strong storm prevailed over the Arctic basin,

which triggered extreme sea ice melt by bringing heat and moisture from the south towards the central Arctic (Parkinson and Comiso, 2013). Other potential trigger of the extreme sea ice melt in 2012, might be a combination of extremely thin sea ice pack and increased upward ocean heat transport, which created conditions that made the sea ice particularly vulnerable to storms (Zhang et al., 2013). The storm in August 2012 allowed a large amount of oceanic heat to be mixed up to the surface, thus enhancing the sea ice melt. Because the

atmosphere is mostly unpredictable beyond 1 or 2 weeks, we were not able to accurately predict, in terms of amplitude, the sea ice conditions that developed because of the Arctic storm in August 2012.

Another challenge for the sea ice community was the predictability of SSIE in 2013. Sea ice extent in September 2013 was characterized by a revival compared to the low values recorded in September 2012 (SSIE in 2013 was 1.69 million square kilometers above the record minimum extent in September 2012). Most of the models, involved in the SIPN, have underestimated the September 2013 sea ice extent, despite the fact that this was not an extreme low sea ice year like 2012. The observed September 2013 sea ice extent lied outside the intervals given with 13 out of 16 predictions, but the modelling methods performed better than the statistical ones (Stroeve et al., 2014). For September 2013, our statistical model performed almost perfectly, giving one of the best predictions (in terms of amplitude) over the validation period. The revival of the sea ice extent in 2013 was due to a combination of different factors: a colder summer over the Arctic basin, compared to 2012, and no storms prevailing throughout the summer months; less winter clouds in January – February 2013, which resulted in more strongly negative surface radiation budget (Liu and Key, 2014); later melt onset, intermittent freezing events and an earlier fall freeze-up (Wang et al., 2015), among others. Summer 2013 was characterized by an unusual low pressure system over much of the Arctic Ocean, which acted as a limiting factor for the heat transport from the south. Both the SLP and air temperature over the Arctic basin were part of our final predictors for the sea ice extent in 2013 (Figure 2 and Figure 3). As such, the accurate predictions based on our statistical model for 2013 may arise from the fact that no extreme weather events were occurring throughout the summer months over the Arctic region. In addition, we had persistent negative temperature anomalies and a long lasting low pressure system prevailing in June and July over the Arctic basin, variables which were used in our forecast model. A high/low skill in the predictability of extreme September sea ice can be the results of extreme spring preconditioning (e.g., very low ice thickness) and/or the results of extremely anomalous summer weather systems, independent of the spring preconditioning. In observation not all extremes are the results of the same forcing, thus implying that different extremes events will have a different level of predictability.

## 5 Conclusions

In this study we have developed a statistical method based on different oceanic and atmospheric variables to estimate the monthly signal and variability of the Arctic sea ice extent. Based on stepwise multi-regression analysis optimal predictors are identified in terms of stability maps to forecast SSIE on pan-Arctic or regional scale. We have demonstrated that our well-established statistical approach can be used as a promising tool to improve the skill of sea ice extent prediction. In the future, the same methodology will be applied to test the potential predictability, up to two years ahead, by taking into account variables with long-term memory (e.g., heat content and water temperature integrated over different depths) for the whole Arctic. For other regions prone to extreme decrease in the sea ice extent (e.g., Chukchi Sea, Beaufort Sea, Barents Sea) as well as for Antarctica the method will also be adopted. Finally, since the concept can be used as an early warning system for September sea ice extent, both at pan-Arctic level as well as regionally, the potential environmental and economic benefits can be very high.

*Acknowledgements*. This study is promoted by Helmholtz funding through the Polar Regions and Coasts in the Changing Earth System (PACES) program of the AWI. Funding by the Helmholtz Climate Initiative REKLIM is gratefully acknowledged. P. Scholz has been funded by the Collaborative Research Centre TRR 181 "Energy Transfer in Atmosphere and Ocean".

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

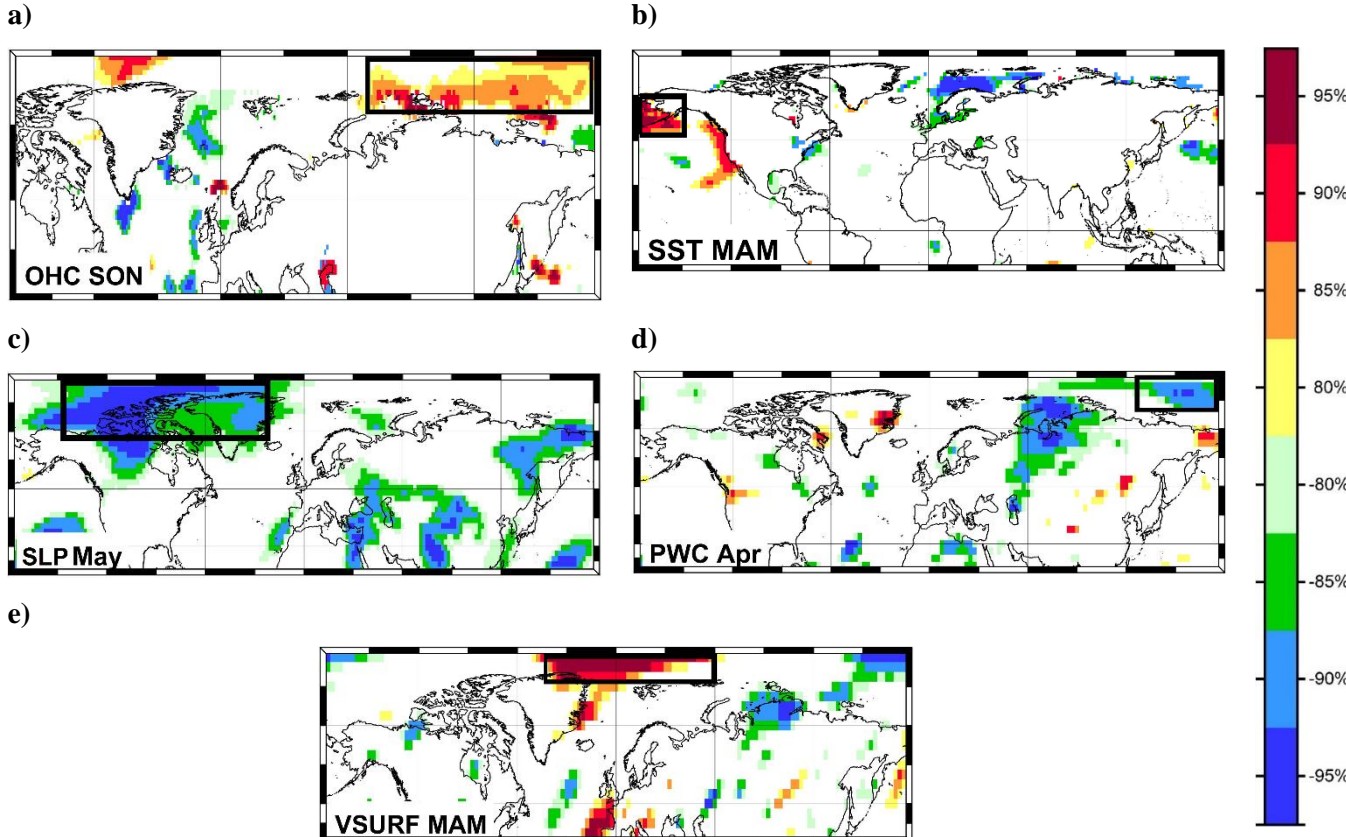

**Figure 1.** Stability map of the correlation between September Sea Ice Extent and a) OHC SON, b) SST MAM, c) SLP May, d) PWC Apr, and e) VSURF MAM. Regions where the correlation is stable, positive and significant for at least 80% of the 21-year windows are shaded with dark red (95%), red (90%), orange (85%) and yellow (80%). The corresponding regions where the correlation is stable, but negative, are shaded with dark blue (95%), blue (90%), green (85%) and light green (80%). The black boxes indicate the regions used for the September sea ice extent at the end of May.

**a)**

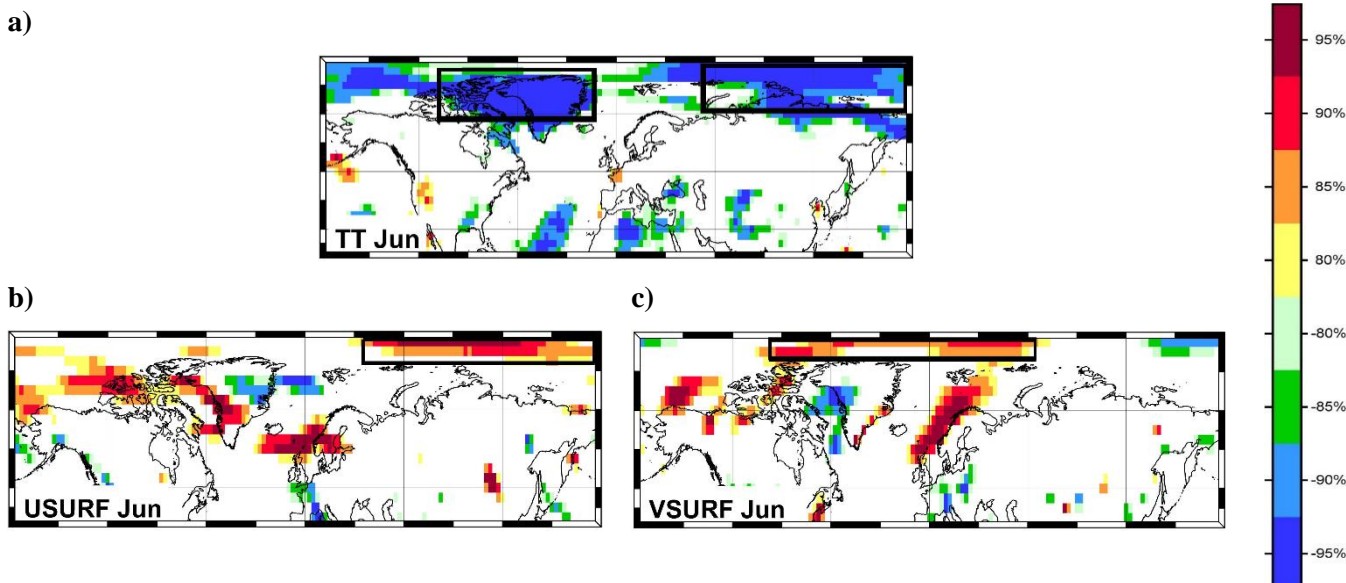

**Figure 2.** Stability map of the correlation between September Sea Ice Extent and a) TT Jun, b) USURF Jun and c) VSURF Jun. Regions where the correlation is stable, positive and significant for at least 80% of the 21-year windows are shaded with dark red (95%), red (90%), orange (85%) and yellow (80%). The corresponding regions where the correlation is stable, but negative, are shaded with dark blue (95%), blue (90%), green (85%) and light green (80%). The black boxes indicate the regions used for the September sea ice extent at the end of June in addition to the variables of May.

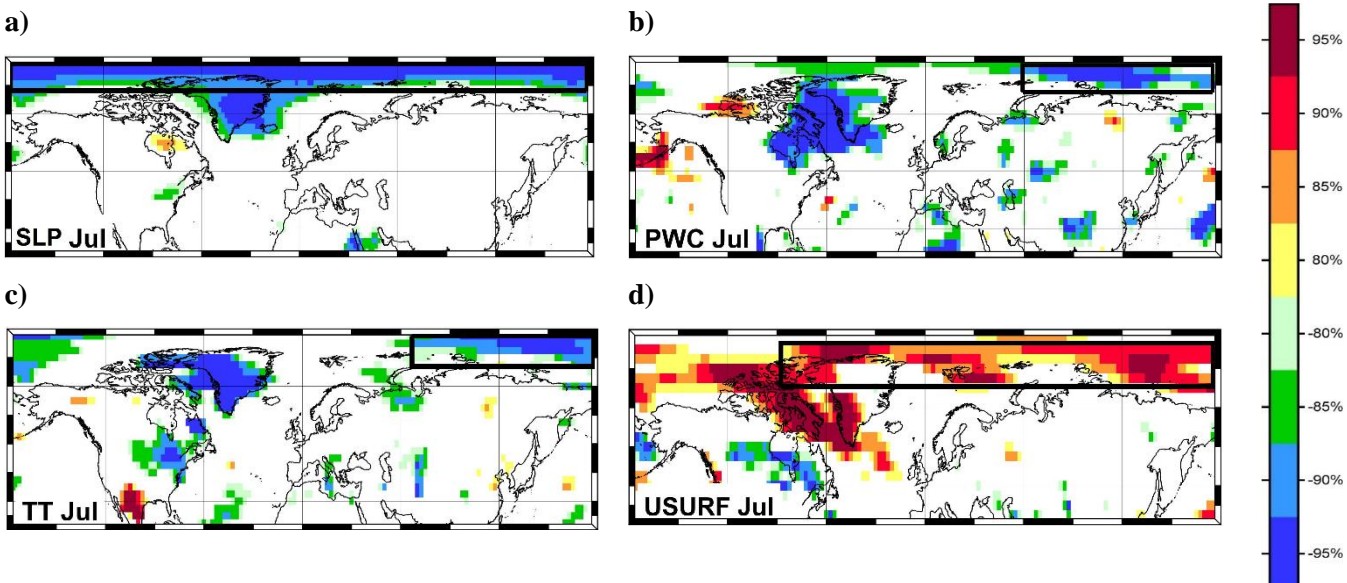

**Figure 3.** Stability map of the correlation between September Sea Ice Extent and a) SLP Jul, b) PWC Jul, c) TT Jul and d) USURF Jul. Regions where the correlation is stable, positive and significant for at least 80% of the 21-year windows are shaded with dark red (95%), red (90%), orange (85%) and yellow (80%). The corresponding regions where the correlation is stable, but negative, are shaded with dark blue (95%), blue (90%), green (85%) and light green (80%). The black boxes indicate the regions used for the September sea ice extent at the end of July in addition to the variables of May and June.

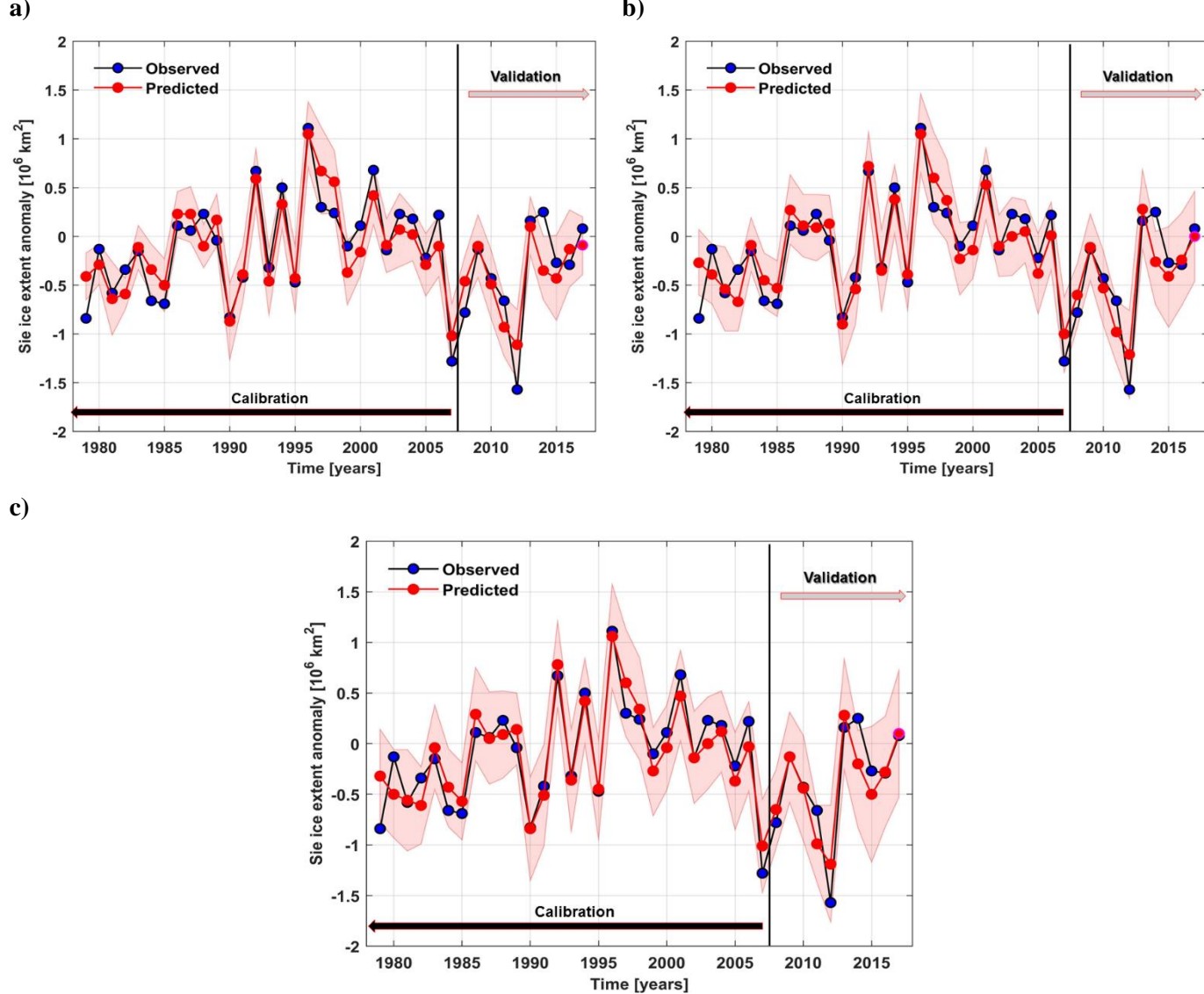

**Figure 4**. Observed (black) and predicted (red) September Sea Ice Extent detrended anomalies over the period 1979-2017 based on a) May, b) June and c) July predictors from the stable regions. The shaded area represents the 95% uncertainty bounds.

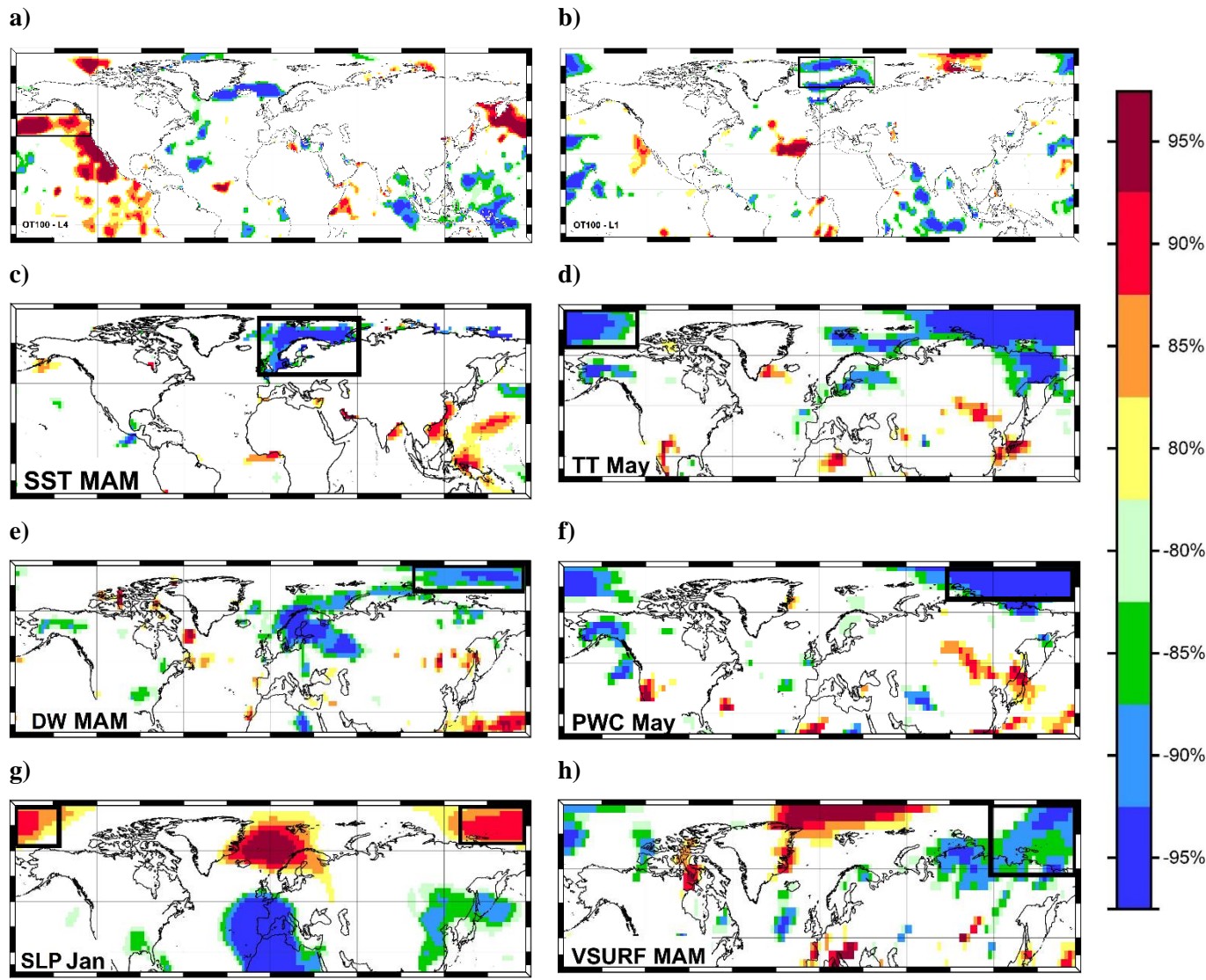

**Figure 5.** Stability map of the correlation between East Siberian September Sea Ice Extent and a) OT100 Annual (L4), b) OT100 Annual (L1), c) SST MAM, d) TT May, e) DW MAM, f) PWC May, g) SLP Jan and h) VSURF MAM. Regions where the correlation is stable, positive and significant for at least 80% of the 21-year windows are shaded with dark red (95%), red (90%), orange (85%) and yellow (80%). The corresponding regions where the correlation is stable, but negative, are shaded with dark blue (95%), blue (90%), green (85%) and light green (80%). The black boxes indicate the regions used for the September sea ice extent at the end of May.

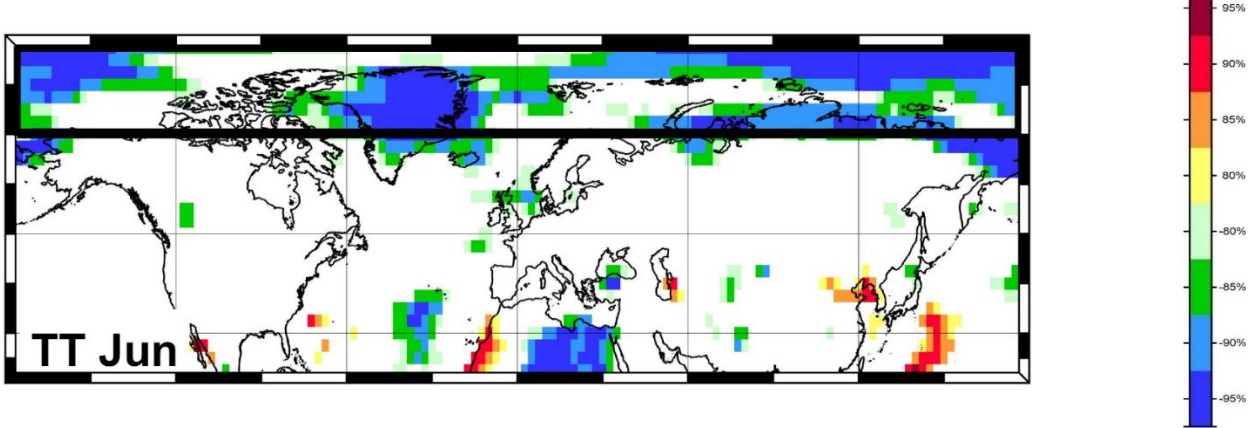

**Figure 6.** Stability map of the correlation between East Siberian September Sea Ice Extent and TT Jun. Regions where the correlation is stable, positive and significant for at least 80% of the 21-year windows are shaded with dark red (95%), red (90%), orange (85%) and yellow (80%). The corresponding regions where the correlation is stable, but negative, are shaded with dark blue (95%), blue (90%), green (85%) and light green (80%). The black boxes indicate the regions used for the September sea ice extent at the end of June in addition to the variables of May.

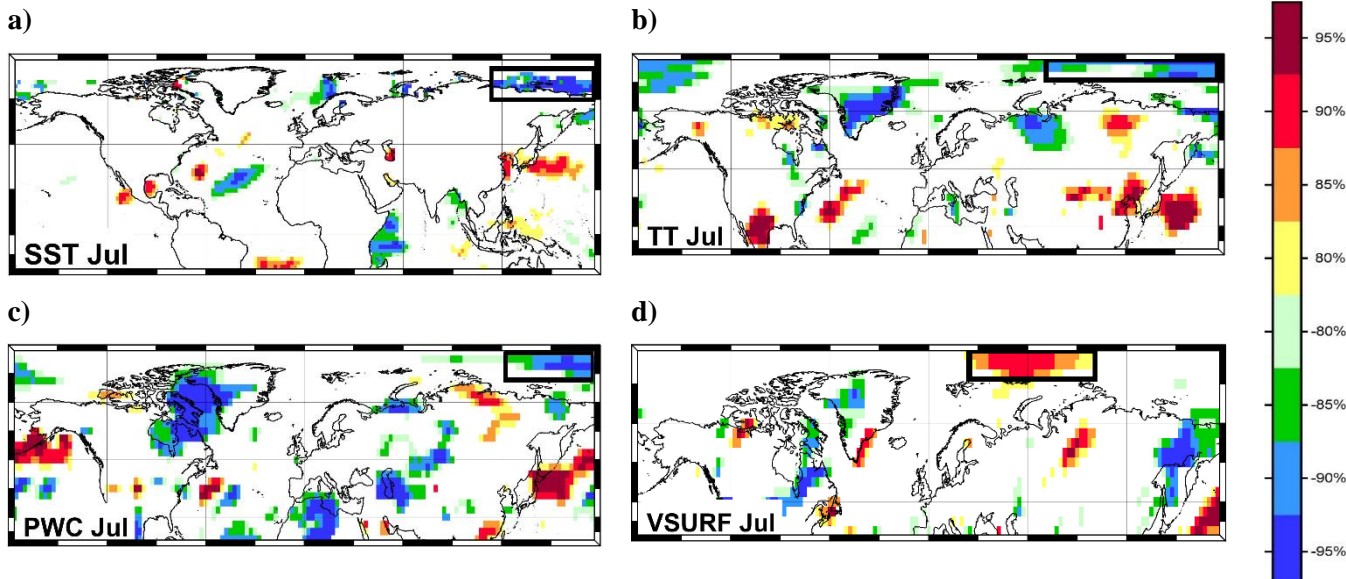

**Figure 7.** Stability map of the correlation between East Siberian September Sea Ice Extent and a) SST Jul, b) TT Jul, c) PWC Jul and d) VSURF Jul. Regions where the correlation is stable, positive and significant for at least 80% of the 21-year windows are shaded with dark red (95%), red (90%), orange (85%) and yellow (80%). The corresponding regions where the correlation is stable, but negative, are shaded with dark blue (95%), blue (90%), green (85%) and light green (80%). The black boxes indicate the regions used for the September sea ice extent at the end of July in addition to the variables of May and June.

a)

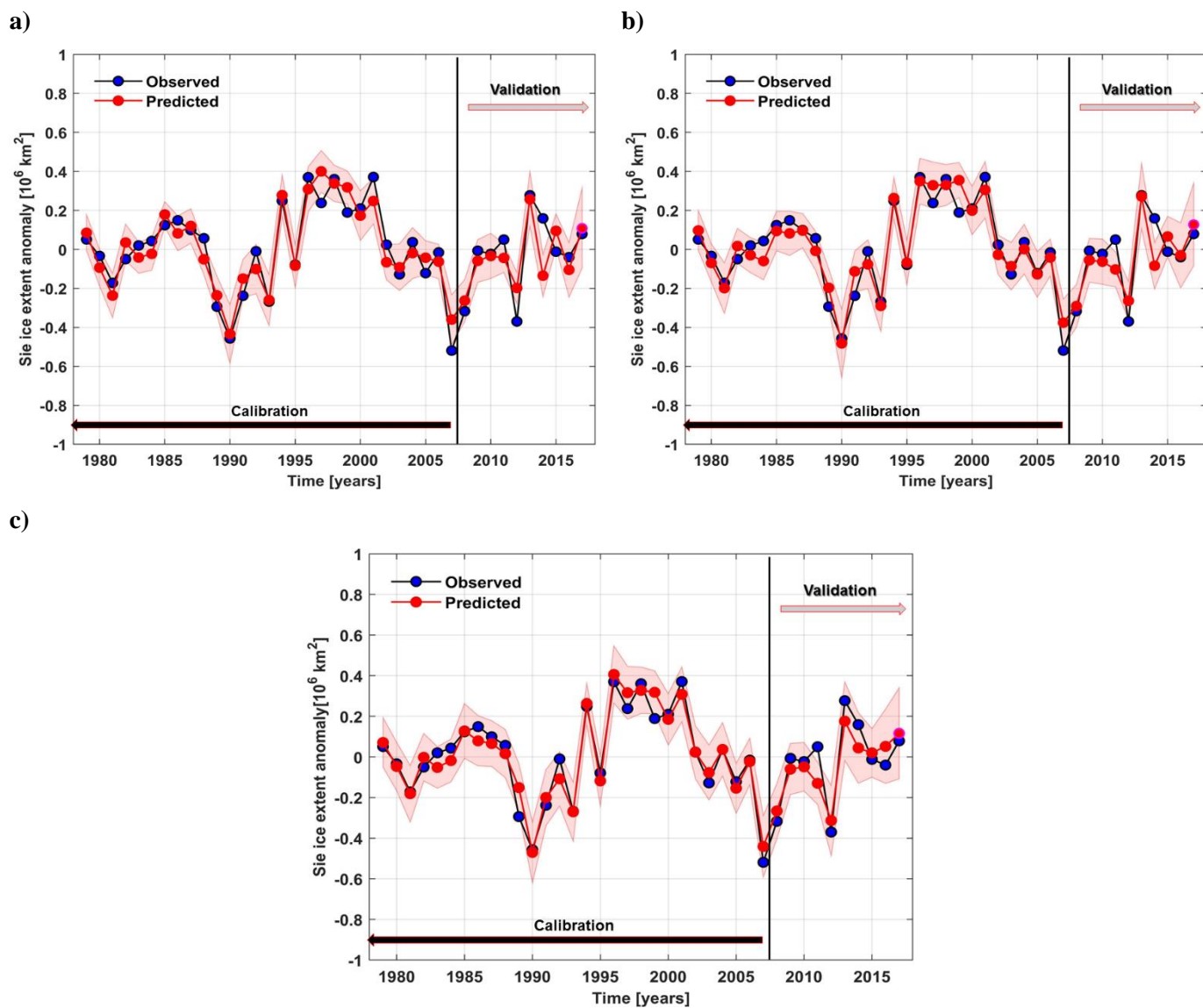

c)

**Figure 8**. Observed (black) and predicted (red) East Siberian Sea Ice Extent detrended anomalies over the period 1979-2017 based on a) May, b) June and c) July predictors from the stable regions. The shaded area represents the 95% uncertainty bounds.

***Table 1***. Name, acronym, source, spatial and temporal resolution of the data sets used in this study

| Name | Source | Temporal resolution | Spatial resolution | Reference |
|---|---|---|---|---|
| Arctic sea ice extent | ftp://sidads.colorado.edu/DATASETS/NOAA/G02135/north/monthly/ | 1979 - 2017 | | Fetterer et al., 2016 |
| AMO index | https://climexp.knmi.nl/data/iamo_ersst.dat | 1979 - 2017 | | Huang et al., 2014 |
| Mean air temperature at 2m (TT) | ftp://ftp.cdc.noaa.gov/Datasets/ncep.reanalysis.derived/surface_gauss/ | 1979 -2017 | 2.5° X 2.5° | Kalnay et al., 1996 |
| Downward longwave radiation (DLR) | ftp://ftp.cdc.noaa.gov/Datasets/ncep.reanalysis.derived/surface_gauss/ | 1979 - 2017 | 2.5° X 2.5° | Kalnay et al., 1996 |
| Zonal surface wind (USURF) | ftp://ftp.cdc.noaa.gov/Datasets/ncep.reanalysis.derived/surface/ | 1979 - 2017 | 2.5° X 2.5° | Kalnay et al., 1996 |
| Meridional surface wind (VSURF) | ftp://ftp.cdc.noaa.gov/Datasets/ncep.reanalysis.derived/surface/ | 1979 - 2017 | 2.5° X 2.5° | Kalnay et al., 1996 |
| Precipitable water content (PWC) | ftp://ftp.cdc.noaa.gov/Datasets/ncep.reanalysis.derived/surface/ | 1979 - 2017 | 2.5° X 2.5° | Kalnay et al., 1996 |
| Sea level pressure (SLP) | ftp://ftp.cdc.noaa.gov/Datasets/ncep.reanalysis.derived/surface/ | 1979 - 2017 | 2.5° X 2.5° | Kalnay et al., 1996 |
| Sea surface temperature (ERSSTv5) | ftp://ftp.ncdc.noaa.gov/pub/data/cmb/ersst/v5/netcdf/ | 1979 - 2017 | 2.0° X 2.0° | Huang et al., 2014 |
| Ocean heat content in the first 700m (OHC) | https://www.nodc.noaa.gov/OC5/3M_HEAT_CONTENT/ | 1979 - 2017 | 2.5° X 2.5° | Levitus et al., 2012 Boyer et al., 2013 |
| Ocean temperature in the first 100m (OT100) | https://www.nodc.noaa.gov/OC5/3M_HEAT_CONTENT/ | 1979 - 2017 | 2.5° X 2.5° | Levitus et al., 2012 Boyer et al., 2013 |

**Table 2.** Time lags used for the forecast of SSIE. Seasonal averages are indicated as winter (December/January/February – DJF), spring (March/April/May – MAM), summer (JJA – June/July/August) and autumn (September/October/December – SON).

| Variable | Time lag | Month | Season |
|---|---|---|---|
| TT, DLR, USURF, VSURF, PWC, SLP | 1 – 7 months, 1 – 2 seasons | January - July | DJF, MAM |
| ERSSTv5 | 1 – 7 months, 1 – 2 seasons | January - July | DJF, MAM |
| OHC, OT100 | 1 – 4 seasons, 1- 4 years | | Annual, DJF, MAM, JJA, SON |
| AMO index | 1- 4 years | | Annual mean |

**Table 3.** Variables retained for the September pan-Arctic sea ice extent forecast (black boxes in Figure 1, 2 and 3). Single months are abbreviated with the first three letters of the month.

| | May Data | June Data | July Data |
|---|---|---|---|
| **Persistence** | SIE Mar | SIE Mar | SIE Mar |
| **Ocean variables** | OHC SON | OHC SON | OHC SON |
| | SST MAM | SST MAM | SST MAM |
| | AMO – L4 | AMO – L4 | AMO – L4 |
| **Atmospheric variables** | SLP May | SLP May | SLP May |
| | | | SLP  Jul |
| | VSURF MAM | VSURF MAM | VSURF MAM |
| | | VSURF Jun | VSURF Jun |
| | | USURF Jun | USURF Jun |
| | | | USURF Jul |
| | PWC Apr | PWC Apr | PWC April |
| | | | PWC Jul |
| | | TT Jun | TT Jun |
| | | | TT Jul |

**Table 4.** The correlation coefficients between the detrended pan-Arctic September sea ice extent and the regional September sea ice extent. A detailed description about the definition of each region is given here: ftp://sidads.colorado.edu/DATASETS/NOAA/G02135/seaice_analysis/

| | Lag 4 | Lag 3 | Lag 2 | Lag 1 | Lag 0 |
|---|---|---|---|---|---|
| **Baffin** | 0.07 | 0.09 | 0.34 | 0.40 | 0.39 |
| **Barents** | 0.20 | 0.16 | 0.27 | 0.13 | 0.14 |
| **Beaufort** | 0.15 | 0.24 | 0.37 | 0.51 | **0.60** |
| **Bering** | -0.30 | -0.02 | 0.14 | 0.00 | -0.04 |
| **Canadian** | 0.07 | -0.16 | 0.01 | 0.52 | 0.49 |
| **Chukchi** | -0.26 | 0.03 | 0.09 | 0.53 | **0.60** |
| **East Siberian** | 0.19 | 0.24 | 0.39 | 0.61 | **0.72** |
| **Greenland** | 0.04 | 0.06 | 0.22 | 0.16 | -0.07 |
| **Hudson** | 0.44 | 0.51 | 0.46 | 0.38 | 0.47 |
| **Kara** | 0.09 | -0.03 | 0.05 | -0.08 | -0.07 |
| **Laptev** | 0.34 | 0.32 | 0.40 | 0.37 | 0.53 |

**Table 5.** Variables retained for the September East Siberian sea (ESS) ice extent forecast (black boxes in Figure 5, 6 and 7). Seasonal averages are indicated as spring MAM (March, April, May); single months are abbreviated with the first three letters of the month.

|  | **May Data** | **June Data** | **July Data** |
|---|---|---|---|
| **Persistence** |  | SIE Jun | SIE Jun |
|  |  |  | SIE Jul |
| **Ocean variables** | OT100 – L4, L1 | OT100 – L4, L1 | OT100 – L4, L1 |
|  | SST MAM | SST MAM | SST MAM |
| **Atmospheric variables** | SLP Jan | SLP Jan | SLP Jan |
|  | VSURF MAM | VSURF MAM | VSURF MAM |
|  |  |  | VSURF Jul |
|  | PWC May | PWC May | PWC May |
|  |  |  | PWC Jul |
|  | TT May | TT May | TT May |
|  |  | TT Jun | TT Jun |
|  |  |  | TT Jul |
|  | DW MAM | DW MAM | DW MAM |

