# Peer review of "September Arctic Sea Ice minimum prediction – a new skillful statistical approach"

_Earth System Dynamics, 2018_

## Referee Comment (RC1) · Anonymous Referee #1 · 29 Oct 2018

The authors use a statistical model to skilfully predict the Arctic September sea ice extent and the regional East Siberian Sea ice extent up to 4 months ahead. They combine several oceanic and atmospheric parameters and sea ice extent itself from previous months and perform a multiple linear regression. Variables and regions are selected based on stable teleconnections between the predictors and the predictand.

This study is an important contribution for seasonal Arctic and regional sea ice predictions. The predicted skill is higher compared with previous studies. The identification of relevant regions and parameters is useful for understanding processes and changes in climate models. I reviewed previous versions of this manuscript and I am pleased with the current version. The focus on de-trended time series and the separation into a calibration and validation period increases the robustness of the results. I strongly

recommend publication and would like to make a few minor comments, only.

Minor comments:

1. Section 2.2: Give reference to stability figures and remove sentence about colours.

2. Section 3.1: Would be nice to get some information about the impact of the individual predictors. It is surprising to see that only March ice extent is used for prediction of pan-Arctic sea ice extent based on June and July data. Is there no additional benefit from April, May and June ice extent?

3. Section 3.2: Would recommend to rename header from "Robustness of the methodology" to "Regional September ice prediction". From Table 2, only Lag 0 results are discussed.

4. Conclusion: "Moreover, our statistical model is able to properly reproduce the years with extreme low / high sea ice extent, both at pan-Arctic level as well as at regional scale (e.g., 2007 and 2012 – low SSIE and 1996 – high SSIE; see Figure 4 and Figure 5)." Given that only 2012 is within the validation period, it is questionable how robust this statement is.

5. Figure 5: Use same legend for all sub-figures.

6. Missing reference: Petty et al. 2017, https://agupubs.onlinelibrary.wiley.com/doi/full/10.1002/2016EF000495

---

## Referee Comment (RC2) · Anonymous Referee #2 · 12 Nov 2018

In this paper, the authors propose a statistical model to predict the Arctic September sea ice extent (SSIE) and East Siberian Sea ice extent (ESSIE) up to 4 months ahead with a high predictive skill compared to previous studies. Stability maps and stepwise multiple regression analysis are applied to find the optimal predictors for the model in regions and variables respectively. The results of prediction here are excellent and reliable. I believe the approach to build statistical prediction model between the predictors and predictand could be widely used in more climate predictions. I recommend to publish this paper and would like to make a few minor comments.

Minor comments:

1. In Figure 1-3 and S1-S3, those regions inside the black boxes are used for SSIE. However, besides those regions, there are also other regions with significant correlation coefficients. Some regions are even more significant than those regions you choose. Why do you only choose those regions in the black boxes? Could you give an explanation?

2. In Section 3.2, I recommend giving the definition or boundary of East Siberian Sea as well as other areas mentioned in Table 2.

3. In Section 3.2, there is a writing mistake in the sentence ". . . coefficient between the observed and forecasted ESS SSIE is r = 0.94 (r = 77) . . .". I think "r = 77" should be corrected as "r = 0.77".

---

## Referee Comment (RC3) · Anonymous Referee #3 · 15 Nov 2018

Review of

September Arctic sea ice minimum prediction - a new skillful statistical approach

by

Ionita, M., et al.

Summary: Multi-annual time series of sea-ice extent and various oceanic and atmospheric parameters are used to establish so-called stability maps. These maps inform about location, degree and significance of linear correlations between sea-ice extent and the other parameters where these correlations are particularly stable over time using a 21-year long time window. Based on the information provided by these maps a step-wise multiple linear regression analysis is carried out to find the optimal param-

eter combination to predice pan-Arctic and Eastern Siberian Sea (ESS) September sea-ice extent (SSIE). Different start months (May to July) are tested. The result of this regression analysis is then used to predict pan-Arctic and ESS SSIE based on the optimal parameter combination. The predicted time series of SSIE are compared to the observed ones. Evaluation of the prediction skill is carried out using various statistical metrics, suggesting that the method has a) high predictive skill and that b) the skill improves the more parameters are used and c) the closer one moves in time to September.

The manuscript might be a useful addition to existing knowledge. However, the lack of detail in description of data and methodology, the lack of transparency in how certain choices in the methodology were made, and the lack of a critical discussion of the methodology and its potential limitations as well as of the results themselves require that a lot more work needs to be put into the manuscript before it can be accepted for publication. It cannot be published in its current form.

In the following I have listed my main concerns in the general comments which are followed by specific comments, which are either a specification of the general comments or point to other weaknesses / ways to improve the manuscript; finally there is a small typos section.

General Comments: GC1: The introduction requires a revision of cited literature which partly seems to be outdated and partly is simply missing.

GC2: The introduction should work out better the choice of the statistical method used and of the atmospheric and oceanographic parameters used. In my view the attempts listed do not provide a clear motivation of the methodology and set of data chosen by you. I suggest to better motivate the choice of method & data by providing more quantitative evidence about which methods failed so far (and why) and which parameters failed so far (and why).

GC3: The description of the used data lacks quite some information and motivation.

It appears that some of the used data sets are outdated. See the respective specific comments.

GC4: The description of the generation of the stability maps should be improved. There are some open questions. See the respective specific comments.

GC5: The description of the stepwise multiple linear regression analysis should be improved. It is very short, some information is missing and, in particular, it is not possible to understand why and how you ended up with the optimal parameters shown in Figures 1 through 3 and Table 1 as well as the supplementary material.

GC6: A discussion of the results is missing completely. Elements could be, for instance: - critical reflections on the method: How physically meaningful are the parameters found to the prediction? Only very briefly mentioned is that a change in the period used for the "calibration" between 15 & 25 years does not make much of a difference. To give this 1-sentence statement a better basis one could show similar figures as Figure 4 using development periods of, e.g., 17 and 25 years (a symmetric change around the chosen 21 years) and discuss how the prediction skill changes and how the improvement in prediction skill from May to July changes. If there is not too much change then this is not necessarily a good sign of the credibility of the method, is it? - critical reflection on the quality of the data used - reflections on the increase in prediction skill the closer one gets to September + an explanation why this increase is more pronounced for the ESS. - reflections on the years where the prediction is particularly good or bad because of which reason? - reflections on your skill metrics: Have others used these metrics to assess sea-ice prediction skill as well already? If yes, which results did they obtain? If not: Which other measures were used and how comparable are your results to those of the others? - reflections in the practical side: Optimal skill is achieved when using a quite large suite of parameters for July, i.e. just two months ahead of the September sea-ice cover minimum. Is this what we aim for? Aren't we after a minimum set of parameters to be used already in May to predict the September sea-ice cover minimum, i.e. four months in advance? The more parameters you

need the more uncertainties (from the larger number of parameters required) might be loaded into your prediction. In that sense it would have been extremely useful to see a discussion about what is the minimum set of (which?) parameters in May to achieve a prediction skill of X.

Specific Comments:

Page 1, Line 23: You include sea-ice thickness here albeit our knowledge about sea-ice thickness decline seems much less solid than our knowledge about sea-ice extent reduction - and to my knowledge there is not yet scientifically proven evidence that the observed sea-ice thickness reduction is also caused by climate change. It is very likely from physical principles that this is the case, but I am not aware that there has been a paper like Notz and Marotzke, 2012, where this issue has been discussed.

Page 1, Lines 27/28: "Grosfeld et al., 2016" –> Albeit I appreciate the activities at AWI to enhance its service with regard to informing scientists and the wider public about topics relevant in polar research, I doubt that in a publication like this manuscript the citation of Grosfeld et al., 2016, should be one of the major citations when it comes to the observed sea-ice decline in the Arctic. Also "Serreze et al., 2007" is about ten years old a citation. Please look for more recent citations of Arctic sea ice decline in the white literature ... Cavalieri and Parkinson, 2012; Comiso et al., 2017; to mention two.

Page 2, Line 4-15: Since your background and motivation to carry out this study seemingly is driven by shipping activities I would have expected a more thorough review of what is available in the literature towards this topic. I would have liked to see, for example, Melia et al., and Pizzolato et al., both 2016, both Geophysical Research Letters; citing the AMAP report is fine (I have it in my shelf as well), but this is from 2004 = almost 15 years old and there have been revisions of this material. In addition, when it comes to potential challenges, then you could have looked into papers such as Lasserre, F., 2014, Polar Record and Larsen, L.-H., et al., 2016, Polar Research and I

bet there are many more papers you could use to support your statements in the last lines of this paragraph.

Page 2, Line 26: Chevallier et al., 2013 and Sigmond et al., 2013 are missing in the reference list.

Page 3, Line 8: This sounds like a valuable approach, however, as you wrote earlier in the introduction, the Arctic system is a very complex system and in the light thereof I find this statement puzzling. It can well be that the correlation between one pair of predictor and predictant changes due to changes in the correlation between another pair which is related to pair one. The influence of one particular parameter to SSIE can grow relative to the influence of another parameter. While correlation with this latter parameter might stay the same, correlation to the former parameter might change (e.g. increase) such that its skill to forecast SSIE increases as well. This approach might therefore a-priori exclude skill-improvements over time.

Page 3, Lines 11-15: This is a fairly larger number of parameters. I am wondering whether, e.g. with the SLP distribution one does not already have enough information about USURF and VSURF since the latter are primarily driven by the pressure gradient (provided) and the air temperature gradient (provided). Or in other words, what is the motivation of your choice of parameters (see GC2).

Page 3, Lines 21-25: - This paragraph lacks information about the time period, grid type, grid resolution, temporal resolution, and satellite from and for which the SIC is obtained. - Please also correct "NSIDC bootstrap algorithm" with "Comiso bootstrap algorithm"; NSIDC is simply the organization hosting and distributing this data. But see further below. - While it is clear from the introduction that you will use certain atmospheric and oceanographic parameters it is not clear why you require SIC and in particular the sea-ice index. This needs to be pointed out either in the introduction or at the beginning of section 2. - Please check whether it is "sea ice extent index" or "sea ice index". - You might also want to mention that the sea-ice index is just one

number per month while in case of the SIC we are talking about maps. - I do not understand why you are using the Comiso Bootstrap algorithm sea-ice concentration from the SIC data set you are citing. If I go to the doi cited behind the Meier et al. 2013 reference then I end up with the NOAA/NSIDC climate data record (CDR) of sea-ice concentrations, version #2. I would like to know why i) you did not use the main CDR, i.e. the combination of Comiso Bootstrap and NASA-Team algorithms, and ii) you did not use version #3 of this data set: https://nsidc.org/data/G02202/versions/3 Please note also, independent of whether you are using #2 or #3 you would need to cite the Peng et al., 2013 paper (see "Citing These Data" under the just above given URL).

Page 3, Line 26-30 / Page 4, Line 1-3: - What is missing here is the time period. For which time period did you actually obtain the reanalysis and also the other data sets listed in this paragraph? - What is your motivation to use a coarse resolution, outdated atmospheric reanalysis when i) there is an updated, finer resolved reanalysis of the same kind available from the same provider: NCEP-DOE and when ii) there are other reanalyses available which might perform better than NCEP/NCAR in the Arctic and the parameters chosen (e.g. JRA-55, ERA-Interim, MERRA-2, etc.)? Please motivate your choice. I guess for your approach it is quite important to have a re-analysis data set which is as accurate and realistic as possible. - What is the advantage of using the ERSST data set over using other, finer resolved SST data sets? While you are using Version #4b there has been a version #5 out for a while, see: Huang, B., Peter W. Thorne, et. al, 2017: Extended Reconstructed Sea Surface Temperature version 5 (ERSSTv5), Upgrades, validations, and intercomparisons. J. Climate, doi: 10.1175/JCLI-D-16-0836.1 Would you mind checking whether version #5 is more accurate in the Arctic Ocean than version #4? I guess it is worth it. It might be good as well to cite a paper or two in which you motivate your choice of using this SST data set over other SST data sets. - Finally, for the global ocean heat content data sets it would also be very good to have a statement pointing towards why this particular data set is chosen.

[Figure]

Page 4, Lines 6-22_ - Line 8/9: Please place the publications related to streamflow predictions behind the "... Danube river)" to ensure easier association of topic and citation. - Line 9/10: When you are referring to "region" then you mean regions where the spatiotemporal distribution of the predictors is stably correlated with the Pan-Arctic SSIE? - Lines 10-12: "from previous months (years)" –> unclear. I assume your time step is a month, because you are using monthly data. Why years? Further, since you did not specify yet the length of the time period you are investigating mentioning of "moving window of 21 years" remains unclear. What is the maximum or minimum time lag with which you compute the correlation? Why 21 years? - Did you carry out any pre-processing before you perform the correlation analysis? From the data section it becomes vaguely clear that the involved data sets potentially have a different grid resolution and are also on different grids. - Is the Student t-test used a two-sided one? - What is the motivation to go down to significance levels of 80%? Usually, researchers are using significance levels of 99% or 95%. - Line 14: "for more than 80% of the 21-year windows" –> It is not entirely clear what you did here. Please be more accurate in the description of how you performed the correlation analysis. If "1979-2007" is the length of your entire data set (I assume now from Line 15) and your moving window is 21 years long, then you have 9 years (=108 months or 9 seasonal cycles) in total for which you compute the correlation. If the correlation is above a certain significance level in 80% of these 9 years, i.e. 7.2 years, then the correlation is considered stable? Why? What is scientific rationale behind this choosing this threshold? - "significance levels that define the stability of the correlation vary within reasonable limits" –> I don't understand this. What are "reasonable limits"? Do you mean that if you are using other significance levels (SL), say 94%, 92%, 87%,and 82% instead of the mentioned ones, doesn't change the result, i.e. the general pattern in the stability maps? Why should it? You are showing a whole suite of SLs already anyways and at the end you take the 90% SL (Line 21) as the one to base you analysis on. I find this statement confusing. I'd rather ask why you did not take a SL of 95% instead of asking whether the stability maps would change with different SL. What is the motivation to vary the length of the

moving window the way you wrote, i.e. 15-25 years? After all, the main contributor to the correlation possibly is the seasonal cycle in the predictors and the predictand. - Lines 21/22: How is the detrending done? Did you compute average seasonal cycles - for each grid cell - for which time period? - It would not hurt to refer to Figure 1 already in this paragraph since you are describing the significance levels in a quite detailed way already.

Page 4, Lines 24-28: - Line 26/27: A good place to again refer to Figure 1.

Page 5, Lines 1-7: - This multiple linear regression is one of the key ingredients of your paper. I therefore suggest that you give more details, like a) How did you technically implement the stepwise linear regression? What is the step size? Are we talking about temporal steps or about steps in terms of parameters used for the linear regression? How is the prioritizing of the predictors quantified? Are the partial correlations in this step wise approach as high as those shown in the stability maps? b) What is the error? How did you derive / quantify the error? c) I assume that Y is dimensionless since you term it an index? Otherwise it has unit square kilometers. d) How is ensured that the equation in Line 3 provides a correct physical unit at the end?

Page 5, Lines 10-25: - This paragraph belongs to either the introduction or the methods section. - Since you have given the citations of the data sets used already in the data set subsection there is not need to repeat them here. - Lines 11/12 vs. lines 14/15: You state that sea-ice cover and snow cover belong to the long-term memory components but then use OHC, OT100 and SST as long-term predictors. Wouldn't it be more straightfoward to only refer to the oceanic components in Lines 11/12 and leave out snow cover and ice cover? - Line 18: "if predictable" –> I don't understand this statement in this context. Why has the atmospheric circulation to be predictable if you aim for the prediction of Septemnber SIE based on spring atmospheric conditions? - Line 21: What do you mean by "advective parameters"? - Lines 22-25: These two sentences should perhaps be re-formulating for clarity and to avoid repetition along the lines: "Atmospheric moisture content, e.g. clouds, water vapor content, has an impact

on the net surface radiation balance and hence also on the SSIE (Kapsch et al., 2013, 2014). As a measure for this impact we use the precipitable water content (PWC) as an additional predictor."

Page 5, Line 26 to Page 6, Line 4: - This paragraph belongs to either the introduction or the methods section. - Line 1: "SSIE index" –> why index? I thought you are after the sea-ice extent? - Lines 2/3: "with different lags, depending on the variable" –> It seems you are using different time lags for different parameters in the correlation analysis? Please specify why. Please detail the time lags associated with which parameter. Line 3/4: "The optimal predictors are defined as the average values over the stable regions for each gridded parameter." –> unclear. Did you take a stability map between, say SSIE and an arbitrary parameter, e.g. SST, and average over the SST within the region defined as having a correlation at 90% significance? And then? Then you have an average SST value ... fine ... and next? Why is this the "optimal predictor"? Further, when doing this averaging, do you take into account the actual correlation value as well or do you only use the significance as a criterion to select over which SST values you are averaging?

Page 6, Lines 5-30: - It is not clear how you end up with the variables and time periods shown in Figures 1 through 3 and Table 1. It is not clear why in particular in some cases optimal predictors either are based on monthly values (of one or even two different months) or are based on seasonal averages; you have not introduces seasonal averages at all yet. This all seems quite arbitrary. - It it not clear what the criterion is to place black boxes in Figures 1 through 3 and what their meaning is. It is not clear in particular how the location of the black boxes go along with the notion that only in these stability maps only regions with > 90% significance are used. This all seems quite arbitrary. - Figures 1 through 3: If the 90% significance level is as important as I think based on what you wrote, then I suggest to mark the 90% also by a change in color in the stability maps. Currently, 90% significance is right in the middle of (regular) red or blue. - In the captions of Figures 2 and 3 you need to state that these
are the ADDITIONAL parameters (additional with respect to May, Figure 1) on which the prediction of the September sea-ice extent is based. - Lines 5/6: What is meant by "...we have retained all the stable regions ... for all variables based on previous months' data."? - Lines 11-22: The entire issue about AMO needs to be put into the methods section where you can introduce this as an important additional parameter. It should be introduced in the RESULTS section.

Figure 4: I suggest to change the y-axis title to "Sea-ice extent anomaly []"; currently it is "Sie ice extent []". This applies also to Figure 5. For Figure 5, I in addition note that the range of the y-axis differs between a) and b)&c); I suggest to make this consistent as there is no physical reason to have differrent ranges. Another comment, maybe a matter of taste, but I would not call your comparison of the last 7 years' skill of the prediction a "validation" - this is an evaluation. I would not call your 21-year period "calibration". Instruments like a pyranometer, pyrgeometer, anemometer, etc., are calibrated. You use that period to develop your method. Hence, I suggest to speak of a "development phase" or "training phase" and an "evaluation phase" It would be good, finally, to also show a graph of the difference between observation and prediction in relative terms; a difference of 300 000 kmˆ2 with respect to the sea-ice extent of the entire Arctic is different in relative terms than a difference of 100 000 kmˆ2 with respect to the ESS sea-ice extent. That way you can better quantify the skill of your method.

Page 7, Lines 1-13: - Line 8: Please make clear that these are again additional parameters, i.e. on top of those for May (Fig. 1) and June (Fig. 2).

Heading of section 3.2: I would reformulate the title of this subsection into: "Application of the methodology for regional SIE prediction" Also, my general view is that you should include all the supplementary figures and table of this subsection into the regular paper.

- Lines 21-25: I don't think that Table 2 and these lines starting with "Moreover, when looking ..." are required to motivate your look at a specific region of the Arctic Ocean.

- A link / hint towards how you ended up at the optimal parameter combination would

be an asset here as well for the correct understanding of your results (see also my comments with respect to Figures 1 through 3 and Table 1 further up).

Page 8, Line 19 3/4: A discussion of the results is missing completely. See GC6

Page 8, Line 21 to Page 9, Line 2: - Line 23: "Although" ... –> Why? - I suggest to move Line 24 - end of this paragraph to a later place in the conclusions. Generally, starting the conclusion with a short summary about what has been done in this study would put whatever information coming later in the conclusion into a better context. - L28-30: I am not sure what the mentioning of the Parkinson et al (2006) paper and the excurse on modelling has to do with your work. After all, inconsistencies between model and observation with regard to sea-ice cover can have MANY reasons. The fact that OHC is one of these is neither new nor would I use that as one of the highlights to show why your study is valuable.

Page 9, Lines 3-24: This is part of the discussion - see GC6. - I suggest to split this paragraph into two in Line 16. the first one could talk about the teleconnections with, e.g. AMO and the impact the AMO has on which variable used in your method - if you find that relevant. Frankly speaking, if Yu et al. (2017) found that a strong link between these large scale pattern - why didn't you use correlations of between sea-ice cover and the strength of these patterns for your predictions? The second one could talk and refer to the regionally / locally driven feedbacks between ocean, sea-ice and atmosphere. You refer to SLP under these more regional forcings (Line 17) - but isn't particularly the SLP distribution the one which is directly linked to PDO, AMO, AO, NAO, etc.?

- Line 4: What do you mean by "climate variables"?

Lines 4-6: - "(the stable regions over the western European coast)" –> I do not find these in your figures. - OHC: For most of the areas shown in Figure 1 a) the correlation with OHC of SON of the previous year (?) is stable with than 90% significance and hence does not enter your prediction according to your description of the method. Does

this change for June & July as starting months of the prediction? - SST: Figure 1b) shows two areas of significant correlation with SST of MAM, a positive one in the Bering Sea and a negative one in the Barents Sea. You only used the one in the Bering Sea. Why?

- Line 10/11/12: Why "anomaly"? - Line 17 ++: I would about the "and so on" and I would try to be as explicit as possible here, referring to the respective Figure(s) for better understanding. It is crucial to discuss these influences in the light of which SLP distribution, associated VSURF and USURF distributions and PWC and TT patterns belong together physically and whether these are reflected in a consistent way in your stability maps. There is a lot to see in these stability maps and potentially also a lot to discuss albeit with the danger that one over-interprets correlations. In that respect you could improve the value of your paper and the discussion / conclusion in particular. A discussion could and should include more information about your choice of regions used for the prediction (black boxes in Figures 1-3). - Line 20-23: You mention PWC and air temperature in Lines 20/21 but then refer to "transport" in Line 23 - so how about the meridional transport of PWC?

Page 9, Line 25 - Page 10, Line 5: - Line 28: 1996 and 2007 lie within your development or training phase while only 2012 lies outside it. Hence one could argue that the only true year for comparison is 2012 for which the agreement is not as good as for 1996, for example. You could include 2013 as well, for which the prediction is quite good. It is further interesting to see that for ESS a start of the prediction in July results in a substantially better agreement for the two years of extreme pan-Arctic minima 2007 and 2012 (compare Fig. 5c with Fig. 5 a). In contrast, for the entire Arctic it does not really make a difference whether you start in May (Fig. 4a) or July (Fig. 4c): the discrepancy between observations and prediction remains unchanged. But this of course belongs to the discussion, in which you need to critically assess your approach to here, in the conclusions, give explicit recommendations about which starting month and which parameters are suited best to achieve the best prediction. Lines 4-5: Yes,

the concept can be used but how much better is it compared to other systems and concept. This you did not demonstrate and/or discuss. Therefore I would delete this last sentence. Another argument for deleting this sentence is that, e.g. shipping, might not be too much interested in the pan-Arctic sea-ice extent. Sea-ice area or, even better, the sea-ice distribution will be a better measure. As an example: The SSIE of the ESS is unchanged compared to winter but the sea-ice area reduced to 1/2 of the winter value if half of the ESS is ice covered by 75% sea ice and half of the ESS is ice covered by 25% sea ice. Hence the value of regional sea-ice extent prediction for shipping remains questionable.

Typos: - Please decide whether you use the American or British way of writing "skillful".

Page 2, L12: "flows" –> "floes"

Page 3, Lines 20 & 24: "extracted". I suggest to write "obtained" or "downloaded".

Page 4 Line 14: "level" –> "significance level" Line 24: "forecast m all" –> "m"? Line 27: "each ... parameters" –> "each ... parameter"

Page 5 Line 18: "substantial" –> "substantially"

Page 7 Line 23: "EES" –> "ESS"

Page 9 Line 11: "EES" –> "ESS" Line 12: "form" –> "from"

---

## Author Comment (AC1) · 18 Dec 2018

The authors use a statistical model to skillfully predict the Arctic September sea ice extent and the regional East Siberian Sea ice extent up to 4 months ahead. They combine several oceanic and atmospheric parameters and sea ice extent itself from previous months and perform a multiple linear regression. Variables and regions are selected based on stable teleconnections between the predictors and the predictand. This study is an important contribution for seasonal Arctic and regional sea ice predictions. The predicted skill is higher compared with previous studies. The identification of relevant regions and parameters is useful for understanding processes and changes in climate models. I reviewed previous versions of this manuscript and I am pleased with the current version. The focus on de-trended time series and the separation into a calibration and validation period increases the robustness of the results. I strongly recommend publication and would like to make a few minor comments, only.

*We thank the reviewer for the comments and useful feedback regarding our manuscript. Please find below our responses to the reviewer's concerns. Comments will carefully be included in the revised version, as they will help to improve the clearness and scientific content.*

Minor comments:
1. Section 2.2: Give reference to stability figures and remove sentence about colors.

*The text will be modified following the aforementioned suggestion.*

2. Section 3.1: Would be nice to get some information about the impact of the individual predictors. It is surprising to see that only March ice extent is used for prediction of pan-Arctic sea ice extent based on June and July data. Is there no additional benefit from April, May and June ice extent?

*A more detailed description regarding the contribution of each parameter will be given in the revised version of the manuscript.*

3. Section 3.2: Would recommend to rename header from "Robustness of the methodology" to "Regional September ice prediction". From Table 2, only Lag 0 results are discussed.

*We will change the title of Section 3.2 from "Robustness of the methodology" to "Regional September ice prediction". Also, we will add information regarding the other lags analyzed to complete the information of Table 2.*

4. Conclusion: "Moreover, our statistical model is able to properly reproduce the years with extreme low / high sea ice extent, both at pan-Arctic level as well as at regional scale (e.g., 2007 and 2012 – low SSIE and 1996 – high SSIE; see Figure 4 and Figure 5)." Given that only 2012 is within the validation period, it is questionable how robust this statement is.

*We agree with the reviewer's comment and we take his recommendation into account and we will modify the text accordingly.*

5. Figure 5: Use same legend for all sub-figures.

*Will be modified as suggested.*

6. Missing reference: Petty et al. 2017,
https://agupubs.onlinelibrary.wiley.com/doi/full/10.1002/2016EF000495

*The reference will be added in the revised version of the manuscript.*

---

## Author Comment (AC2) · 18 Dec 2018

In this paper, the authors propose a statistical model to predict the Arctic September sea ice extent (SSIE) and East Siberian Sea ice extent (ESSIE) up to 4 months ahead with a high predictive skill compared to previous studies. Stability maps and stepwise multiple regression analysis are applied to find the optimal predictors for the model in regions and variables respectively. The results of prediction here are excellent and reliable. I believe the approach to build statistical prediction model between the predictors and predictand could be widely used in more climate predictions. I recommend to publish this paper and would like to make a few minor comments.

*We thank the reviewer for the comments and useful feedback regarding our manuscript. Please find below our responses to the reviewer's concerns. Comments will carefully be included in the revised version, as they will help to improve the clearness and scientific content.*

Minor comments:
1. In Figure 1-3 and S1-S3, those regions inside the black boxes are used for SSIE. However, besides those regions, there are also other regions with significant correlation coefficients. Some regions are even more significant than those regions you choose. Why do you only choose those regions in the black boxes? Could you give an explanation?

*An explanation why do we only choose those regions in the black boxes will be added in the revised version of the manuscript.*

2. In Section 3.2, I recommend giving the definition or boundary of East Siberian Sea as well as other areas mentioned in Table 2.

*A figure with the regions mentioned in the manuscript, as well as a clear definition of the regions, will be included in the revised version of the manuscript.*

3. In Section 3.2, there is a writing mistake in the sentence ". . . coefficient between the observed and forecasted ESS SSIE is r = 0.94 (r = 77) . . .". I think "r = 77" should be corrected as "r = 0.77".

*The text will be modified accordingly.*

---

## Author Comment (AC3) · 18 Dec 2018

Review of September Arctic sea ice minimum prediction - a new skillful statistical approach
By Ionita, M., et al.

Summary: Multi-annual time series of sea-ice extent and various oceanic and atmospheric parameters are used to establish so-called stability maps. These maps inform about location, degree and significance of linear correlations between sea-ice extent and the other parameters where these correlations are particularly stable over time using a 21-year long time window. Based on the information provided by these maps a step-wise multiple linear regression analysis is carried out to find the optimal parameter combination to predict pan-Arctic and Eastern Siberian Sea (ESS) September sea-ice extent (SSIE). Different start months (May to July) are tested. The result of this regression analysis is then used to predict pan-Arctic and ESS SSIE based on the optimal parameter combination. The predicted time series of SSIE are compared to the observed ones. Evaluation of the prediction skill is carried out using various statistical metrics, suggesting that the method has a) high predictive skill and that b) the skill improves the more parameters are used and c) the closer one moves in time to September.

The manuscript might be a useful addition to existing knowledge. However, the lack of detail in description of data and methodology, the lack of transparency in how certain choices in the methodology were made, and the lack of a critical discussion of the methodology and its potential limitations as well as of the results themselves require that a lot more work needs to be put into the manuscript before it can be accepted for publication. It cannot be published in its current form. In the following I have listed my main concerns in the general comments which are followed by specific comments, which are either a specification of the general comments or point to other weaknesses / ways to improve the manuscript; finally there is a small typos section.

*We thank the reviewer for the comments and useful feedback regarding our manuscript. Please find below our responses to the reviewer's concerns. We outline how we will deal some of the major comments when revising the paper. We tried to answer those detailed and thoroughly including references to underline our arguments. Comments will carefully be included in the revised version, as they will help to improve the clearness and scientific content.*

General Comments:

GC1: The introduction requires a revision of cited literature which partly seems to be outdated and partly is simply missing.

*We will consider this aspect carefully and check at each reference place if there are newer references available. If so, we will include them in order to update the scientific state of the art discussion.*

GC2: The introduction should work out better the choice of the statistical method used and of the atmospheric and oceanographic parameters used. In my view the attempts listed do not provide a clear motivation of the methodology and set of data chosen by you. I suggest to better motivate the choice of method & data by providing more quantitative evidence about which methods failed so far (and why) and which parameters failed so far (and why).

*Introduction starts with an overview of how Arctic sea ice is changing and which role an accurate prediction of the sea ice situation plays in terms of economics. Then we turn to the current possibilities of predicting the sea ice situation. This is followed by a paragraph discussing the different approaches how they perform and which are there limitations. This leads to the final paragraph where we highlight the motivation and why we try to come along with the above-*

*mentioned challenges by using a statistical approach that is well documented in the literature. Unfortunately, these references where not included in the actual manuscript. We will include them in the revised version. By this we can emphasize why the method performed is based on good statistical practice and we can demonstrate the high expertise we have in regard to the statistical method we are presenting here. The same statistical approach is used in pre-operational mode for the streamflow prediction for Elbe and Rhine streamflow (Ionita et al., 2009, 2014, 2017; Meißner et al., 2017) and we strongly feel that the method as well as our results have both the physical and statistical meaning to be published in the context of sea ice prediction.*

GC3: The description of the used data lacks quite some information and motivation. It appears that some of the used data sets are outdated. See the respective specific comments.

*We comment on this below in the specific comments in order to avoid any repetition.*

GC4: The description of the generation of the stability maps should be improved. There are some open questions. See the respective specific comments.

*We comment on this below in the specific comments in order to avoid any repetition.*

GC5: The description of the stepwise multiple linear regression analysis should be improved. It is very short, some information is missing and, in particular, it is not possible to understand why and how you ended up with the optimal parameters shown in Figures 1 through 3 and Table 1 as well as the supplementary material.

*We agree with the reviewer that this paragraph is rather short. We will revise this by adding some additional information and some references to underline our approach.*

GC6: A discussion of the results is missing completely. Elements could be, for instance: - critical reflections on the method: How physically meaningful are the parameters found to the prediction? Only very briefly mentioned is that a change in the period used for the "calibration" between 15 & 25 years does not make much of a difference. To give this 1-sentence statement a better basis one could show similar figures as Figure 4 using development periods of, e.g., 17 and 25 years (a symmetric change around the chosen 21 years) and discuss how the prediction skill changes and how the improvement in prediction skill from May to July changes. If there is not too much change then this is not necessarily a good sign of the credibility of the method, is it? - critical reflection on the quality of the data used - reflections on the increase in prediction skill the closer one gets to September + an explanation why this increase is more pronounced for the ESS. - reflections on the years where the prediction is particularly good or bad because of which reason? - reflections on your skill metrics: Have others used these metrics to assess sea-ice prediction skill as well already? If yes, which results did they obtain? If not: Which other measures were used and how comparable are your results to those of the others? - reflections in the practical side: Optimal skill is achieved when using a quite large suite of parameters for July, i.e. just two months ahead of the September sea-ice cover minimum. Is this what we aim for? Aren't we after a minimum set of parameters to be used already in May to predict the September sea-ice cover minimum, i.e. four months in advance? The more parameters you need the more uncertainties (from the larger number of parameters required) might be loaded into your prediction. In that sense it would have been extremely useful to see a discussion about what is the minimum set of (which?) parameters in May to achieve a prediction skill of X.

*By reading the comments and once again the above mentioned chapters it seems clear that they need some more structure to help the reader following our analysis, obtained results and the discussion on it. We will also add some subtitles to split text in meaningful subsections. We believe that this changes combined with working on the specific comments, will answer many of the comments made in GC6.*

**Specific Comments:**

Page 1, Line 23: You include sea-ice thickness here albeit our knowledge about sea ice thickness decline seems much less solid than our knowledge about sea-ice extent reduction - and to my knowledge there is not yet scientifically proven evidence that the observed sea-ice thickness reduction is also caused by climate change. It is very likely from physical principles that this is the case, but I am not aware that there has been a paper like Notz and Marotzke, 2012, where this issue has been discussed.

*Following the reviewer's comments, we can change the sentence as follow to be more precise here: The sea ice extent over the Arctic has undergone an extraordinary decline during the last decades that can be linked to climate change (Allison et al., 2009; Kay et al., 2011; Notz and Marotzke, 2012). Observations of sea ice thickness help us to understand the Arctic climate, and have the potential to support seasonal forecasts and operational activities in the polar regions.*

Page 1, Lines 27/28: "Grosfeld et al., 2016" –> Albeit I appreciate the activities at AWI to enhance its service with regard to informing scientists and the wider public about topics relevant in polar research, I doubt that in a publication like this manuscript the citation of Grosfeld et al., 2016, should be one of the major citations when it comes to the observed sea-ice decline in the Arctic. Also "Serreze et al., 2007" is about ten years old a citation. Please look for more recent citations of Arctic sea ice decline in the white literature ... Cavalieri and Parkinson, 2012; Comiso et al., 2017; to mention two.

*We will follow the reviewer's suggestion and add some more recent references in this part of the manuscript.*

Page 2, Line 4-15: Since your background and motivation to carry out this study seemingly is driven by shipping activities I would have expected a more thorough review of what is available in the literature towards this topic. I would have liked to see, for example, Melia et al., and Pizzolato et al., both 2016, both Geophysical Research Letters; citing the AMAP report is fine (I have it in my shelf as well), but this is from 2004 = almost 15 years old and there have been revisions of this material. In addition, when it comes to potential challenges, then you could have looked into papers such as Lasserre, F., 2014, Polar Record and Larsen, L.-H., et al., 2016, Polar Research and I bet there are many more papers you could use to support your statements in the last lines of this paragraph.

*We will add some more recent references on this part.*

Page 2, Line 26: Chevallier et al., 2013 and Sigmond et al., 2013 are missing in the reference list.

*We will add these references in the reference list.*

Page 3, Line 8: This sounds like a valuable approach, however, as you wrote earlier in the introduction, the Arctic system is a very complex system and in the light thereof I find this statement puzzling. It can well be that the correlation between one pair of predictor and predictand changes due to changes in the correlation between another pair which is related to pair one. The influence of one particular parameter to SSIE can grow relative to the influence of another parameter. While correlation with this latter parameter might stay the same, correlation to the former parameter might change (e.g. increase) such that its skill to forecast SSIE increases as well. This approach might therefore a-priori exclude skill-improvements over time.

*Although we agree with this comment, we have to keep in mind that each model (statistical or dynamical) offers room for improvement. In this paper, we want to show a different statistical approach, compared to the existing ones, and its potential advantages. By using running correlations, we try to avoid, at least partially, the non-stationarity issue between different variables. As mentioned above the same statistical approach is used in pre-operational mode for the streamflow prediction for Elbe and Rhine streamflow (Ionita et al., 2009, 2014, 2017; Meißner et al., 2017) and we strongly feel that the method as well as our results have both the*

*physical and statistical meaning required to work in context of sea ice prediction. The method was also used to investigate the "Moisture transport and Antarctic sea ice: austral spring 2016 event" (Ionita et. Al. 2018).*

Page 3, Lines 11-15: This is a fairly larger number of parameters. I am wondering whether, e.g. with the SLP distribution one does not already have enough information about USURF and VSURF since the latter are primarily driven by the pressure gradient (provided) and the air temperature gradient (provided). Or in other words, what is the motivation of your choice of parameters (see GC2).

*If one looks at the stability maps, you can see that the stable regions are located over different regions for SLP, USURF and VSURF. In some cases, SLP is not even selected as a potential predictor. It is true that SLP, USURF and VSURF are interconnected, but in the current case, the stable regions are independent of each other, thus we choose to use all there variables.*

Page 3, Lines 21-25: - This paragraph lacks information about the time period, grid type, grid resolution, temporal resolution, and satellite from and for which the SIC is obtained. - Please also correct "NSIDC bootstrap algorithm" with "Comiso bootstrap algorithm"; NSIDC is simply the organization hosting and distributing this data. But see further below. - While it is clear from the introduction that you will use certain atmospheric and oceanographic parameters it is not clear why you require SIC and in particular the sea-ice index. This needs to be pointed out either in the introduction or at the beginning of section 2. - Please check whether it is "sea ice extent index" or "sea ice index". - You might also want to mention that the sea-ice index is just one number per month while in case of the SIC we are talking about maps. - I do not understand why you are using the Comiso Bootstrap algorithm sea-ice concentration from the SIC data set you are citing. If I go to the doi cited behind the Meier et al. 2013 reference then I end up with the NOAA/NSIDC climate data record (CDR) of sea-ice concentrations, version #2. I would like to know why i) you did not use the main CDR, i.e. the combination of Comiso Bootstrap and NASA-Team algorithms, and ii) you did not use version #3 of this data set: https://nsidc.org/data/G02202/versions/3 Please note also, independent of whether you are using #2 or #3 you would need to cite the Peng et al., 2013 paper (see "Citing These Data" under the just above given URL).

*We are thankful for the reviewer comments to be more precisely on describing the data base of sea ice which will be implemented into the manuscript.*

Page 3, Line 26-30 / Page 4, Line 1-3: - What is missing here is the time period. For which time period did you actually obtain the reanalysis and also the other data sets listed in this paragraph? - What is your motivation to use a coarse resolution, outdated atmospheric reanalysis when i) there is an updated, finer resolved reanalysis of the same kind available from the same provider: NCEP-DOE and when ii) there are other reanalyses available which might perform better than NCEP/NCAR in the Arctic and the parameters chosen (e.g. JRA-55, ERA-Interim, MERRA-2, etc.)? Please motivate your choice. I guess for your approach it is quite important to have a re-analysis data set which is as accurate and realistic as possible. - What is the advantage of using the ERSST data set over using other, finer resolved SST data sets? While you are using Version #4b there has been a version #5 out for a while, see: Huang, B., Peter W. Thorne, et. al, 2017: Extended Reconstructed Sea Surface Temperature version 5 (ERSSTv5), Upgrades, validations, and intercomparisons. J. Climate, doi: 10.1175/JCLI-D-16-0836.1 Would you mind checking whether version #5 is more accurate in the Arctic Ocean than version #4? I guess it is worth it. It might be good as well to cite a paper or two in which you motivate your choice of using this SST data set over other SST data sets. - Finally, for the global ocean heat content data sets it would also be very good to have a statement pointing towards why this particular data set is chosen.

*Again we are thankful for the reviewer comments on our data base. We will in general add the time periods used for making the analysis. Also we will add sentences to explain our choice. We have to mention more clear that the choice of our datasets is based on their availability. Although ERA-interim would be a better option compared to NCEP data, ERA-interim is not updated in real time, thus making impossible to be used in "operational" mode. We are participating in the Sea Ice Prediction Network forecast challenge, thus we need the data to be publically available at the*

*beginning of each month. We agree that it's optimal to have the proper reanalysis dataset, but we have to take into account also their availability in near real time. Regarding the ERSST data, there was a mistake from our side. We are using the ERSSTv5 for the forecast, not the previous version. We will correct this n the revised version of the manuscript. To make it easier for the reader, we will add a table with all the datasets used in the manuscript, their proper references and the link where the data can be downloaded.*

Page 4, Lines 6-22_ - Line 8/9: Please place the publications related to streamflow predictions behind the "... Danube river)" to ensure easier association of topic and citation. –

*References will be added in the revised version of the manuscript.*

Line 9/10: When you are referring to "region" then you mean regions where the spatiotemporal distribution of the predictors is stably correlated with the Pan-Arctic SSIE?

*In the revised version, we will clearly define "region" in the context of our analysis.*

Lines 10-12: "from previous months (years)" –> unclear. I assume your time step is a month, because you are using monthly data. Why years? Further, since you did not specify yet the length of the time period you are investigating mentioning of "moving window of 21 years" remains unclear. What is the maximum or minimum time lag with which you compute the correlation? Why 21 years? - Did you carry out any pre-processing before you perform the correlation analysis? From the data section it becomes vaguely clear that the involved data sets potentially have a different grid resolution and are also on different grids. - Is the Student t-test used a two-sided one? - What is the motivation to go down to significance levels of 80%? Usually, researchers are using significance levels of 99% or 95%.

*This will be written more clearly in the revised version of the manuscript. We will describe the time period and we will refer to the references where the method is used in order to underline the used moving window of 21 years. Sentence will be split in order to help the reader following the approach.*

Line 14: "for more than 80% of the 21-year windows" –> It is not entirely clear what you did here. Please be more accurate in the description of how you performed the correlation analysis. If "1979-2007" is the length of your entire data set (I assume now from Line 15) and your moving window is 21 years long, then you have 9 years (=108 months or 9 seasonal cycles) in total for which you compute the correlation. If the correlation is above a certain significance level in 80% of these 9 years, i.e. 7.2 years, then the correlation is considered stable? Why? What is scientific rationale behind this choosing this threshold? - "significance levels that define the stability of the correlation vary within reasonable limits" –> I don't understand this. What are "reasonable limits"? Do you mean that if you are using other significance levels (SL), say 94%, 92%, 87%,and 82% instead of the mentioned ones, doesn't change the result, i.e. the general pattern in the stability maps? Why should it? You are showing a whole suite of SLs already anyways and at the end you take the 90% SL (Line 21) as the one to base you analysis on. I find this statement confusing. I'd rather ask why you did not take a SL of 95% instead of asking whether the stability maps would change with different SL. What is the motivation to vary the length of the moving window the way you wrote, i.e. 15-25 years? After all, the main contributor to the correlation possibly is the seasonal cycle in the predictors and the predictand.

*We think that parts of this comment are  not applicable (e.g. the use of 94%, 92%, 87%,and 82% significance level) . We use a well-established and published methodology. For the final predictors we select the regions where the correlation is stable above the 95% significance level. Everything below this level is not taken into account and it's used like a buffer zone. Nevertheless, this paragraph will be overworked and re-written so that the reader can follow the approach. By this, we are convinced that the reviewers comments are fulfilled.*

Lines 21/22: How is the detrending done? Did you compute average seasonal cycles - for each grid cell - for which time period? - It would not hurt to refer to Figure 1 already in this paragraph since you are describing the significance levels in a quite detailed way already.

*We will describe what we did in terms of detrending in the revised version of the paper.*

Page 4, Lines 24-28: - Line 26/27: A good place to again refer to Figure 1.

*We will change the text accordingly.*

Page 5, Lines 1-7: - This multiple linear regression is one of the key ingredients of your paper. I therefore suggest that you give more details, like a) How did you technically implement the stepwise linear regression? What is the step size? Are we talking about temporal steps or about steps in terms of parameters used for the linear regression? How is the prioritizing of the predictors quantified? Are the partial correlations in this step wise approach as high as those shown in the stability maps? b) What is the error? How did you derive / quantify the error? c) I assume that Y is dimensionless since you term it an index? Otherwise it has unit square kilometers. d) How is ensured that the equation in Line 3 provides a correct physical unit at the end?

*This paragraph will be overworked and re-written so that the reader can follow the approach. By this, we are convinced that the reviewers comments are fulfilled.*

Page 5, Lines 10-25: - This paragraph belongs to either the introduction or the methods section. - Since you have given the citations of the data sets used already in the data set subsection there is not need to repeat them here.

*We agree with the reviewer and we will shift the paragraph to the data section.*

Lines 11/12 vs. lines 14/15: You state that sea-ice cover and snow cover belong to the long-term memory components but then use OHC, OT100 and SST as long-term predictors. Wouldn't it be more straightforward to only refer to the oceanic components in Lines 11/12 and leave out snow cover and ice cover?

*The text will be modified following the reviewer's suggestions.*

Line 18: "if predictable" –> I don't understand this statement in this context. Why has the atmospheric circulation to be predictable if you aim for the prediction of September SIE based on spring atmospheric conditions?

*We will delete "if predictable" in the revised version of the paper.*

Line 21: What do you mean by "advective parameters"? - Lines 22-25: These two sentences should perhaps be re-formulating for clarity and to avoid repetition along the lines: "Atmospheric moisture content, e.g. clouds, water vapor content, has an impact on the net surface radiation balance and hence also on the SSIE (Kapsch et al., 2013, 2014). As a measure for this impact we use the precipitable water content (PWC) as an additional predictor."

*We will rephrase these sentences following the suggestion of the reviewer in the revised manuscript.*

Page 5, Line 26 to Page 6, Line 4: - This paragraph belongs to either the introduction or the methods section.

*We agree with the reviewer and we will shift the paragraph to the data section.*

Line 1: "SSIE index" –> why index? I thought you are after the sea-ice extent? - Lines 2/3: "with different lags, depending on the variable" –> It seems you are using different time lags for different parameters in the correlation analysis? Please specify why. Please detail the time lags associated with which parameter. Line 3/4: "The optimal predictors are defined as the average values over the stable regions for each gridded parameter." –> unclear. Did you take a stability map between, say SSIE and an arbitrary parameter, e.g. SST, and average over the SST within the region defined as having a correlation at 90% significance? And then? Then you have an average SST value ... fine ... and next? Why is this the "optimal predictor"? Further, when doing this averaging, do you take into account the actual correlation value as well or do you only use the significance as a criterion to select over which SST values you are averaging?

*This paragraph will be overworked and re-written so that the reader can follow the approach. By this, we are convinced that the reviewers comments are fulfilled.*

Page 6, Lines 5-30: - It is not clear how you end up with the variables and time periods shown in Figures 1 through 3 and Table 1. It is not clear why in particular in some cases optimal predictors either are based on monthly values (of one or even two different months) or are based on seasonal averages; you have not introduces seasonal averages at all yet. This all seems quite arbitrary. - It it not clear what the criterion is to place black boxes in Figures 1 through 3 and what their meaning is. It is not clear in particular how the location of the black boxes go along with the notion that only in these stability maps only regions with > 90% significance are used. This all seems quite arbitrary. - Figures 1 through 3: If the 90% significance level is as important as I think based on what you wrote, then I suggest to mark the 90% also by a change in color in the stability maps. Currently, 90% significance is right in the middle of (regular) red or blue. - In the captions of Figures 2 and 3 you need to state that these are the ADDITIONAL parameters (additional with respect to May, Figure 1) on which the prediction of the September sea-ice extent is based. - Lines 5/6: What is meant by "...we have retained all the stable regions ... for all variables based on previous months' data."?

*The method and approach is not all "arbitrary" as mentioned by the reviewer. Nevertheless, we think the reviewer is correct that we have to present the paragraph more structured and clearer concerning the analysis. For this, we will rephrase sentences and make the steps clearer. Furthermore, the figure captions will get all information asked for to understand them better. Not all this call into question the principal approach of the method itself.*

Lines 11-22: The entire issue about AMO needs to be put into the methods section where you can introduce this as an important additional parameter. It should be introduced in the RESULTS section.

*A more detailed discussion regarding the influence of AMO on the Arctic sea ice variability will be added in the revised version of our manuscript.*

Figure 4: I suggest to change the y-axis title to "Sea-ice extent anomaly []"; currently it is "Sie ice extent []". This applies also to Figure 5. For Figure 5, I in addition note that the range of the y-axis differs between a) and b)&c); I suggest to make this consistent as there is no physical reason to have different ranges. Another comment, maybe a matter of taste, but I would not call your comparison of the last 7 years' skill of the prediction a "validation" - this is an evaluation. I would not call your 21-year period "calibration". Instruments like a pyranometer, pyrgeometer, anemometer, etc., are calibrated. You use that period to develop your method. Hence, I suggest to speak of a "development phase" or "training phase" and an "evaluation phase" It would be good, finally, to also show a graph of the difference between observation and prediction in relative terms; a difference of 300 000 km^2 with respect to the sea-ice extent of the entire Arctic is different in relative terms than a difference of 100 000 km^2 with respect to the ESS sea-ice extent. That way you can better quantify the skill of your method.

*We are thankful of the reviewers comments on Figure 4 as by sure it is the anomaly depicted here. This will be changed in the revised version of the manuscript. As well, the ranges will be equalized in order to make the figures directly comparable. Regarding the "calibration" and*

*"validation" wording, this is the state of the art wording in forecasting. All the papers mentioned in the references, and not only, use the terms "calibration" and "validation" to train and evaluate their model.*

Page 7, Lines 1-13: - Line 8: Please make clear that these are again additional parameters, i.e. on top of those for May (Fig. 1) and June (Fig. 2).

*We will add some sentences towards this in the revised version of the manuscript.*

Heading of section 3.2: I would reformulate the title of this subsection into: "Application of the methodology for regional SIE prediction" Also, my general view is that you should include all the supplementary figures and table of this subsection into the regular paper.

*As mentioned in GC6 we think we will work on titles and restructure parts of the manuscript. Following the reviewer's suggestions most of the supplementary figures will be included in the regular manuscript.*

Lines 21-25: I don't think that Table 2 and these lines starting with "Moreover, when looking ..." are required to motivate your look at a specific region of the Arctic Ocean. A link / hint towards how you ended up at the optimal parameter combination would be an asset here as well for the correct understanding of your results (see also my comments with respect to Figures 1 through 3 and Table 1 further up).

*The sentence can be rephrased to match with the reviewer comments.*

Page 8, Line 19 3/4: A discussion of the results is missing completely. See GC6

*See our response below GC8.*

Page 8, Line 21 to Page 9, Line 2: - Line 23: "Although" ... –> Why? - I suggest to move Line 24 - end of this paragraph to a later place in the conclusions. Generally, starting the conclusion with a short summary about what has been done in this study would put whatever information coming later in the conclusion into a better context.

*This can be easily taken into account in the revised version of the manuscript.*

L28-30: I am not sure what the mentioning of the Parkinson et al (2006) paper and the excurse on modelling has to do with your work. After all, inconsistencies between model and observation with regard to sea-ice cover can have MANY reasons. The fact that OHC is one of these is neither new nor would I use that as one of the highlights to show why your study is valuable.

*This reference and the text associated with it will be removed from the revised version of the manuscript.*

Page 9, Lines 3-24: This is part of the discussion - see GC6. - I suggest to split this paragraph into two in Line 16. the first one could talk about the teleconnections with, e.g. AMO and the impact the AMO has on which variable used in your method – if you find that relevant. Frankly speaking, if Yu et al. (2017) found that a strong link between these large scale pattern - why didn't you use correlations of between sea-ice cover and the strength of these patterns for your predictions? The second one could talk and refer to the regionally / locally driven feedbacks between ocean, sea-ice and atmosphere. You refer to SLP under these more regional forcings (Line 17) - but isn't particularly the SLP distribution the one which is directly linked to PDO, AMO, AO, NAO, etc.?

*We follow the suggestion of the reviewer and will make the paragraph clearer. Parts of it will be shifted to a new "Discussion" paragraph.*

Line 4: What do you mean by "climate variables"?

*A sentence will be add to clarify this.*

Lines 4-6: - "(the stable regions over the western European coast)" –> I do not find these in your figures. - OHC: For most of the areas shown in Figure 1 a) the correlation with OHC of SON of the previous year (?) is stable with than 90% significance and hence does not enter your prediction according to your description of the method. Does this change for June & July as starting months of the prediction? - SST: Figure 1b) shows two areas of significant correlation with SST of MAM, a positive one in the Bering Sea and a negative one in the Barents Sea. You only used the one in the Bering Sea. Why?

*This will be rephrased to make the results more understandable. It will not change the major outcomes of the analysis.*

Line 10/11/12: Why "anomaly"? - Line 17 ++: I would about the "and so on" and I would try to be as explicit as possible here, referring to the respective Figure(s) for better understanding. It is crucial to discuss these influences in the light of which SLP distribution, associated VSURF and USURF distributions and PWC and TT patterns belong together physically and whether these are reflected in a consistent way in your stability maps. There is a lot to see in these stability maps and potentially also a lot to discuss albeit with the danger that one over-interprets correlations. In that respect you could improve the value of your paper and the discussion / conclusion in particular. A discussion could and should include more information about your choice of regions used for the prediction (black boxes in Figures 1-3).

*This comment of the reviewer will be fulfilled when adding a section "Discussion" in the paper. By this the suggested refer to the figures can easily be included and the analysis and results will be better to follow.*

Line 20-23: You mention PWC and air temperature in Lines 20/21 but then refer to "transport" in Line 23 - so how about the meridional transport of PWC?

*Sentences will be add to clarify this point.*

Page 9, Line 25 - Page 10, Line 5: - Line 28: 1996 and 2007 lie within your development or training phase while only 2012 lies outside it. Hence one could argue that the only true year for comparison is 2012 for which the agreement is not as good as for 1996, for example. You could include 2013 as well, for which the prediction is quite good. It is further interesting to see that for ESS a start of the prediction in July results in a substantially better agreement for the two years of extreme pan-Arctic minima 2007 and 2012 (compare Fig. 5c with Fig. 5 a). In contrast, for the entire Arctic it does not really make a difference whether you start in May (Fig. 4a) or July (Fig. 4c): the discrepancy between observations and prediction remains unchanged. But this of course belongs to the discussion, in which you need to critically assess your approach to here, in the conclusions, give explicit recommendations about which starting month and which parameters are suited best to achieve the best prediction. Lines 4-5: Yes, the concept can be used but how much better is it compared to other systems and concept. This you did not demonstrate and/or discuss. Therefore I would delete this last sentence. Another argument for deleting this sentence is that, e.g. shipping, might not be too much interested in the pan-Arctic sea-ice extent. Sea-ice area or, even better, the sea-ice distribution will be a better measure. As an example: The SSIE of the ESS is unchanged compared to winter but the sea-ice area reduced to 1/2 of the winter value if half of the ESS is ice covered by 75% sea ice and half of the ESS is ice covered by 25% sea ice. Hence the value of regional sea-ice extent prediction for shipping remains questionable.

*By adding a new paragraph "Discussion", these comments of the reviewer will be addressed. Last sentence can be rephrased.*

Typos: - Please decide whether you use the American or British way of writing "skillful".
Page 2, L12: "flows" –> "floes"
Page 3, Lines 20 & 24: "extracted". I suggest to write "obtained" or "downloaded".
Page 4 Line 14: "level" –> "significance level" Line 24: "forecast m all" –> "m"? Line 27: "each ... parameters" –> "each ... parameter"
Page 5 Line 18: "substantial" –> "substantially"
Page 7 Line 23: "EES" –> "ESS"
Page 9 Line 11: "EES" –> "ESS" Line 12: "form" –> "from"

**_This will all be changed in the revised version of the manuscript._**

---

## Author Response (AR1)

The authors use a statistical model to skillfully predict the Arctic September sea ice extent and the regional East Siberian Sea ice extent up to 4 months ahead. They combine several oceanic and atmospheric parameters and sea ice extent itself from previous months and perform a multiple linear regression. Variables and regions are selected based on stable teleconnections between the predictors and the predictand. This study is an important contribution for seasonal Arctic and regional sea ice predictions. The predicted skill is higher compared with previous studies. The identification of relevant regions and parameters is useful for understanding processes and changes in climate models. I reviewed previous versions of this manuscript and I am pleased with the current version. The focus on de-trended time series and the separation into a calibration and validation period increases the robustness of the results. I strongly recommend publication and would like to make a few minor comments, only.

*We thank the reviewer for the comments and useful feedback regarding our manuscript. Please find below our responses to the reviewer's concerns. Comments will carefully be included in the revised version, as they will help to improve the clearness and scientific content.*

Minor comments:
1. Section 2.2: Give reference to stability figures and remove sentence about colors.

*The text has been modified following the aforementioned suggestion.*

2. Section 3.1: Would be nice to get some information about the impact of the individual predictors. It is surprising to see that only March ice extent is used for prediction of pan-Arctic sea ice extent based on June and July data. Is there no additional benefit from April, May and June ice extent?

*The revised version of the manuscript has been substantially modified trying to take into account all the reviewers suggestions.*

3. Section 3.2: Would recommend to rename header from "Robustness of the methodology" to "Regional September ice prediction". From Table 2, only Lag 0 results are discussed.

*The title of Section 3.2 has been renamed from "Robustness of the methodology" to "Application of the methodology for regional SSIE prediction". We discussed just the Lag 0 because our aim was to identify the region which shows the highest correlation with the pan-Arctic sea ice extent in September.*

4. Conclusion: "Moreover, our statistical model is able to properly reproduce the years with extreme low / high sea ice extent, both at pan-Arctic level as well as at regional scale (e.g., 2007 and 2012 – low SSIE and 1996 – high SSIE; see Figure 4 and Figure 5)." Given that only 2012 is within the validation period, it is questionable how robust this statement is.

*We agree with the reviewer's comment and we took his recommendation into account and we have modified the text accordingly.*

5. Figure 5: Use same legend for all sub-figures.

*Modified as suggested.*

6. Missing reference: Petty et al. 2017,
https://agupubs.onlinelibrary.wiley.com/doi/full/10.1002/2016EF000495

*The reference has been added in the revised version of the manuscript.*
In this paper, the authors propose a statistical model to predict the Arctic September sea ice extent (SSIE) and East Siberian Sea ice extent (ESSIE) up to 4 months ahead with a high predictive skill compared to previous studies. Stability maps and stepwise multiple regression analysis are applied to find the optimal predictors for the model in regions and variables respectively. The results of prediction here are excellent and reliable. I believe the approach to build statistical prediction model between the predictors and predictand could be widely used in more climate predictions. I recommend to publish this paper and would like to make a few minor comments.

*We thank the reviewer for the comments and useful feedback regarding our manuscript. Please find below our responses to the reviewer's concerns. Comments will carefully be included in the revised version, as they will help to improve the clearness and scientific content.*

Minor comments:
1. In Figure 1-3 and S1-S3, those regions inside the black boxes are used for SSIE. However, besides those regions, there are also other regions with significant correlation coefficients. Some regions are even more significant than those regions you choose. Why do you only choose those regions in the black boxes? Could you give an explanation?

*An explanation why do we only choose those regions in the black boxes has been added in the revised version of the manuscript.*

2. In Section 3.2, I recommend giving the definition or boundary of East Siberian Sea as well as other areas mentioned in Table 2.

*Modified as suggested.*

3. In Section 3.2, there is a writing mistake in the sentence ". . . coefficient between the observed and forecasted ESS SSIE is r = 0.94 (r = 77) . . .". I think "r = 77" should be corrected as "r = 0.77".

*Modified as suggested.*
Review of September Arctic sea ice minimum prediction - a new skillful statistical approach
By Ionita, M., et al.

Summary: Multi-annual time series of sea-ice extent and various oceanic and atmospheric parameters are used to establish so-called stability maps. These maps inform about location, degree and significance of linear correlations between sea-ice extent and the other parameters where these correlations are particularly stable over time using a 21-year long time window. Based on the information provided by these maps a step-wise multiple linear regression analysis is carried out to find the optimal parameter combination to predict pan-Arctic and Eastern Siberian Sea (ESS) September sea-ice extent (SSIE). Different start months (May to July) are tested. The result of this regression analysis is then used to predict pan-Arctic and ESS SSIE based on the optimal parameter combination. The predicted time series of SSIE are compared to the observed ones. Evaluation of the prediction skill is carried out using various statistical metrics, suggesting that the method has a) high predictive skill and that b) the skill improves the more parameters are used and c) the closer one moves in time to September.

The manuscript might be a useful addition to existing knowledge. However, the lack of detail in description of data and methodology, the lack of transparency in how certain choices in the methodology were made, and the lack of a critical discussion of the methodology and its potential limitations as well as of the results themselves require that a lot more work needs to be put into the manuscript before it can be accepted for publication. It cannot be published in its current form. In the following I have listed my main concerns in the general comments which are followed by specific comments, which are either a specification of the general comments or point to other weaknesses / ways to improve the manuscript; finally there is a small typos section.

*We thank the reviewer for the comments and useful feedback regarding our manuscript. Please find below our responses to the reviewer's concerns. We have tried to answer those detailed and thoroughly including references to underline our arguments. Comments have been carefully included in the revised version, as they helps us to improve the clearness and the scientific content.*

General Comments:

GC1: The introduction requires a revision of cited literature which partly seems to be outdated and partly is simply missing.

*The introduction part has been modified and up to date references have been added.*

GC2: The introduction should work out better the choice of the statistical method used and of the atmospheric and oceanographic parameters used. In my view the attempts listed do not provide a clear motivation of the methodology and set of data chosen by you. I suggest to better motivate the choice of method & data by providing more quantitative evidence about which methods failed so far (and why) and which parameters failed so far (and why).

*Introduction part has been modified according the reviewers suggestions. Therefore, the introduction starts with an overview of how Arctic sea ice is changing and which role an accurate prediction of the sea ice situation plays in terms of economics. Then we turn to the current possibilities of predicting the sea ice situation. This is followed by a paragraph discussing the different approaches how they perform and which are there limitations. This leads to the final paragraph where we highlight the motivation and why we try to come along with the above-mentioned challenges by using a statistical approach that is well documented in the literature. Unfortunately, these references where not included in the submitted manuscript. We included them in the revised version. By this we can emphasize why the method performed is based on a state of the art and good statistical practice and*

*we can demonstrate the high expertise we have in regard to the statistical method we are presenting here. The same statistical approach is used in pre-operational mode for the streamflow prediction for Elbe and Rhine streamflow (Ionita et al., 2009, 2014, 2017; Meißner et al., 2017) and we strongly feel that the method as well as our results have both the physical and statistical meaning to be published in the context of sea ice prediction.*
*We have to mention that the aim of this paper is to show the advantages of our statistical approach. It is meant in a more technical direction. We agree that the psychical mechanism behind our stable regions have to be properly described and we have tried to do so in the revised version, but we have to keep in mind that a full overview for each region would require a much more longer manuscript and it is beyond the scope of our paper.*

GC3: The description of the used data lacks quite some information and motivation. It appears that some of the used data sets are outdated. See the respective specific comments.

*We comment on this below in the specific comments in order to avoid any repetition.*

GC4: The description of the generation of the stability maps should be improved. There are some open questions. See the respective specific comments.

*We comment on this below in the specific comments in order to avoid any repetition.*

GC5: The description of the stepwise multiple linear regression analysis should be improved. It is very short, some information is missing and, in particular, it is not possible to understand why and how you ended up with the optimal parameters shown in Figures 1 through 3 and Table 1 as well as the supplementary material.

*The description of the multiple linear regression has been improved in the revised version of the manuscript (see Methods section).*

GC6: A discussion of the results is missing completely. Elements could be, for instance: - critical reflections on the method: How physically meaningful are the parameters found to the prediction? Only very briefly mentioned is that a change in the period used for the "calibration" between 15 & 25 years does not make much of a difference. To give this 1-sentence statement a better basis one could show similar figures as Figure 4 using development periods of, e.g., 17 and 25 years (a symmetric change around the chosen 21 years) and discuss how the prediction skill changes and how the improvement in prediction skill from May to July changes. If there is not too much change then this is not necessarily a good sign of the credibility of the method, is it? - critical reflection on the quality of the data used - reflections on the increase in prediction skill the closer one gets to September + an explanation why this increase is more pronounced for the ESS. - reflections on the years where the prediction is particularly good or bad because of which reason? - reflections on your skill metrics: Have others used these metrics to assess sea-ice prediction skill as well already? If yes, which results did they obtain? If not: Which other measures were used and how comparable are your results to those of the others? - reflections in the practical side: Optimal skill is achieved when using a quite large suite of parameters for July, i.e. just two months ahead of the September sea-ice cover minimum. Is this what we aim for? Aren't we after a minimum set of parameters to be used already in May to predict the September sea-ice cover minimum, i.e. four months in advance? The more parameters you need the more uncertainties (from the larger number of parameters required) might be loaded into your prediction. In that sense it would have been extremely useful to see a discussion about what is the minimum set of (which?) parameters in May to achieve a prediction skill of X.

*Following the reviewer's suggestion we have modified the manuscript substantially, especially the introduction and the conclusions part, in order to take into account the aforementioned concerns.*

**Specific Comments:**
Page 1, Line 23: You include sea-ice thickness here albeit our knowledge about sea ice thickness decline seems much less solid than our knowledge about sea-ice extent reduction - and to my knowledge there is not yet scientifically proven evidence that the observed sea-ice thickness reduction is also caused by climate change. It is very likely from physical principles that this is the case, but I am not aware that there has been a paper like Notz and Marotzke, 2012, where this issue has been discussed.

*We agree with the reviewer's concern, thus we have removed "sea ice thickness" from the text.*

Page 1, Lines 27/28: "Grosfeld et al., 2016" –> Albeit I appreciate the activities at AWI to enhance its service with regard to informing scientists and the wider public about topics relevant in polar research, I doubt that in a publication like this manuscript the citation of Grosfeld et al., 2016, should be one of the major citations when it comes to the observed sea-ice decline in the Arctic. Also "Serreze et al., 2007" is about ten years old a citation. Please look for more recent citations of Arctic sea ice decline in the white literature ... Cavalieri and Parkinson, 2012; Comiso et al., 2017; to mention two.

*We followed the reviewer's suggestion and added more recent references in this part of the manuscript.*

Page 2, Line 4-15: Since your background and motivation to carry out this study seemingly is driven by shipping activities I would have expected a more thorough review of what is available in the literature towards this topic. I would have liked to see, for example, Melia et al., and Pizzolato et al., both 2016, both Geophysical Research Letters; citing the AMAP report is fine (I have it in my shelf as well), but this is from 2004 = almost 15 years old and there have been revisions of this material. In addition, when it comes to potential challenges, then you could have looked into papers such as Lasserre, F., 2014, Polar Record and Larsen, L.-H., et al., 2016, Polar Research and I bet there are many more papers you could use to support your statements in the last lines of this paragraph.

*We have changed to introduction to take into account the aforementioned suggestions and references.*

Page 2, Line 26: Chevallier et al., 2013 and Sigmond et al., 2013 are missing in the reference list.

*We added these references in the reference list.*

Page 3, Line 8: This sounds like a valuable approach, however, as you wrote earlier in the introduction, the Arctic system is a very complex system and in the light thereof I find this statement puzzling. It can well be that the correlation between one pair of predictor and predictand changes due to changes in the correlation between another pair which is related to pair one. The influence of one particular parameter to SSIE can grow relative to the influence of another parameter. While correlation with this latter parameter might stay the same, correlation to the former parameter might change (e.g. increase) such that its skill to forecast SSIE increases as well. This approach might therefore a-priori exclude skill-improvements over time.

*Although we agree with this comment, we have to keep in mind that each model (statistical or dynamical) offers room for improvement. In this paper, we want to show a different statistical approach, compared to the existing ones, and its potential advantages. By using running correlations, we try to avoid, at least partially, the non-stationarity issue between different variables. As mentioned above the same statistical approach is used in pre-operational mode for the streamflow prediction for Elbe and Rhine streamflow (Ionita et al., 2009, 2014, 2017; Meißner et al., 2017) and we strongly feel that the method as well as our results have both the physical and statistical meaning required to work in context of sea ice prediction. The method was also used to investigate the "Moisture transport and Antarctic sea ice: austral spring 2016 event" (Ionita et. al. 2018).*

Page 3, Lines 11-15: This is a fairly larger number of parameters. I am wondering whether, e.g. with the SLP distribution one does not already have enough information about USURF and VSURF since the latter are primarily driven by the pressure gradient (provided) and the air temperature gradient (provided). Or in other words, what is the motivation of your choice of parameters (see GC2).

*If one looks at the stability maps, you can see that the stable regions are located over different regions for SLP, USURF and VSURF. In some cases, SLP is not even selected as a potential predictor. It is true that SLP, USURF and VSURF are interconnected, but in the current case, the stable regions are independent of each other, thus we choose to use all there variables.*

Page 3, Lines 21-25: - This paragraph lacks information about the time period, grid type, grid resolution, temporal resolution, and satellite from and for which the SIC is obtained. - Please also correct "NSIDC bootstrap algorithm" with "Comiso bootstrap algorithm"; NSIDC is simply the organization hosting and distributing this data. But see further below. - While it is clear from the introduction that you will use certain atmospheric and oceanographic parameters it is not clear why you require SIC and in particular the sea-ice index. This needs to be pointed out either in the introduction or at the beginning of section 2. - Please check whether it is "sea ice extent index" or "sea ice index". - You might also want to mention that the sea-ice index is just one number per month while in case of the SIC we are talking about maps. - I do not understand why you are using the Comiso Bootstrap algorithm sea-ice concentration from the SIC data set you are citing. If I go to the doi cited behind the Meier et al. 2013 reference then I end up with the NOAA/NSIDC climate data record (CDR) of sea-ice concentrations, version #2. I would like to know why i) you did not use the main CDR, i.e. the combination of Comiso Bootstrap and NASA-Team algorithms, and ii) you did not use version #3 of this data set: https://nsidc.org/data/G02202/versions/3 Please note also, independent of whether you are using #2 or #3 you would need to cite the Peng et al., 2013 paper (see "Citing These Data" under the just above given URL).

*The data section has been modified following the reviewer's comment. For the forecast purposes in our manuscript we are using just the Arctic sea ice extent index. Thus, the other data have been removed from the revised version. We are not using the sea ice concentration, so we have deleted all the information about this part. The revised version of the manuscript has included a new Table 1, which summarizes all data sets used with source, temporal and spatial resolution as well as the references. Therefore, we think that the reader can easily catch up with the database of the analysis performed in this study.*

Page 3, Line 26-30 / Page 4, Line 1-3: - What is missing here is the time period. For which time period did you actually obtain the reanalysis and also the other data sets listed in this paragraph? - What is your motivation to use a coarse resolution, outdated atmospheric reanalysis when i) there is an updated, finer resolved reanalysis of the same kind available from the same provider: NCEP-DOE and when ii) there are other reanalyses available which might perform better than NCEP/NCAR in the Arctic and the parameters chosen (e.g. JRA-55, ERA-Interim, MERRA-2, etc.)? Please motivate your choice. I guess for your approach it is quite important to have a re-analysis data set which is as accurate and realistic as possible. - What is the advantage of using the ERSST data set over using other, finer resolved SST data sets? While you are using Version #4b there has been a version #5 out for a while, see: Huang, B., Peter W. Thorne, et. al, 2017: Extended Reconstructed Sea Surface Temperature version 5 (ERSSTv5), Upgrades, validations, and intercomparisons. J. Climate, doi: 10.1175/JCLI-D-16-0836.1 Would you mind checking whether version #5 is more accurate in the Arctic Ocean than version #4? I guess it is worth it. It might be good as well to cite a paper or two in which you motivate your choice of using this SST data set over other SST data sets. - Finally, for the global ocean heat content data sets it would also be very good to have a statement pointing towards why this particular data set is chosen.

*Again we are thankful for the reviewer comments on our data base. We have to point out that the choice of our datasets is mainly based on their availability. Although ERA-interim (HadSST) would be a better option compared to NCEP (ERSST) data, ERA-interim (HadSST) unfortunately are not updated in real time, thus making it impossible to be used in "operational" mode. We are participating in the Sea Ice Prediction Network forecast*

*challenge, thus we need the data to be publically available at the beginning of each month. We agree that it's optimal to have the proper reanalysis dataset, but we have to take into account also their availability in near real time. Regarding the ERSST data, there was a written mistake from our side. We have of course used the last version of the ERSST data for the forecast, namely the ERSSTv5, not the previous version. We have corrected this in the revised version of the manuscript. To make it easier for the reader, we also added an overview table (Table 1), listing all the datasets used in this study, their proper references and the link where the data can be downloaded.*

Page 4, Lines 6-22_ - Line 8/9: Please place the publications related to streamflow predictions behind the "... Danube river)" to ensure easier association of topic and citation. –

*Modified as suggested.*

Line 9/10: When you are referring to "region" then you mean regions where the spatiotemporal distribution of the predictors is stably correlated with the Pan-Arctic SSIE?

*The text has been modified accordingly to answer this question.*

Lines 10-12: "from previous months (years)" –> unclear. I assume your time step is a month, because you are using monthly data. Why years? Further, since you did not specify yet the length of the time period you are investigating mentioning of "moving window of 21 years" remains unclear. What is the maximum or minimum time lag with which you compute the correlation? Why 21 years? - Did you carry out any pre-processing before you perform the correlation analysis? From the data section it becomes vaguely clear that the involved data sets potentially have a different grid resolution and are also on different grids. - Is the Student t-test used a two-sided one? - What is the motivation to go down to significance levels of 80%? Usually, researchers are using significance levels of 99% or 95%.

*This section was rephrased in the revised version of the manuscript to clarify the questions raised. We described the time periods and we referred to the references where the method is used in order to underline the used moving window of 21 years. We have added also a table (table 2) in which we show the time lags used for each variable. The grid of each data set is now mentioned in the new table 1. For the statistical significance we used a two-sided student t-test – we have inserted a small paragraph stating this in the methods section. As we already mentioned we used 80% and 85% level as buffer zones. The data from these regions are not taken into account in our forecast exercise.*

Line 14: "for more than 80% of the 21-year windows" –> It is not entirely clear what you did here. Please be more accurate in the description of how you performed the correlation analysis. If "1979-2007" is the length of your entire data set (I assume now from Line 15) and your moving window is 21 years long, then you have 9 years (=108 months or 9 seasonal cycles) in total for which you compute the correlation. If the correlation is above a certain significance level in 80% of these 9 years, i.e. 7.2 years, then the correlation is considered stable? Why? What is scientific rationale behind this choosing this threshold? - "significance levels that define the stability of the correlation vary within reasonable limits" –> I don't understand this. What are "reasonable limits"? Do you mean that if you are using other significance levels (SL), say 94%, 92%, 87%,and 82% instead of the mentioned ones, doesn't change the result, i.e. the general pattern in the stability maps? Why should it? You are showing a whole suite of SLs already anyways and at the end you take the 90% SL (Line 21) as the one to base you analysis on. I find this statement confusing. I'd rather ask why you did not take a SL of 95% instead of asking whether the stability maps would change with different SL. What is the motivation to vary the length of the moving window the way you wrote, i.e. 15-25 years? After all, the main contributor to the correlation possibly is the seasonal cycle in the predictors and the predictand.

*We think concerning our method and expertise that parts of this comment are not applicable (e.g. the use of 94%, 92%, 87%,and 82% significance level). We use a well-established and published methodology. For the final predictors we select the regions where the correlation is stable above the 90% significance level. Everything below this level*

*is not taken into account and it's used like a buffer zone. Although the length of our time series is relatively short (40 years) the methodology proved to work also in cases of timer series <40 years (Ionita et al., 2018). Moreover, we use the same methodology, with the same number of years (40 years), for the prediction of September Arctic and Antarctic Sea Ice (https://www.arcus.org/sipn/sea-ice-outlook/2017/post-season). We varied the length on the moving window, just to be sure that the spatial structure of the stability maps remains the same. We choose the period 1979-2007 as calibration period, as both extreme years of sea ice extent, namely 1996 and 2007, were included and it provides a climate relevant period of nearly 30 years.*

Lines 21/22: How is the detrending done? Did you compute average seasonal cycles - for each grid cell - for which time period? - It would not hurt to refer to Figure 1 already in this paragraph since you are describing the significance levels in a quite detailed way already.

*The detrending procedure has been included in the revised version of the manuscript in the section data and methods. We also make reference to Figure 1 in the description of the stability maps procedure.*

Page 4, Lines 24-28: - Line 26/27: A good place to again refer to Figure 1.

*Modified as suggested.*

Page 5, Lines 1-7: - This multiple linear regression is one of the key ingredients of your paper. I therefore suggest that you give more details, like a) How did you technically implement the stepwise linear regression? What is the step size? Are we talking about temporal steps or about steps in terms of parameters used for the linear regression? How is the prioritizing of the predictors quantified? Are the partial correlations in this step wise approach as high as those shown in the stability maps? b) What is the error? How did you derive / quantify the error? c) I assume that Y is dimensionless since you term it an index? Otherwise it has unit square kilometers. d) How is ensured that the equation in Line 3 provides a correct physical unit at the end?

*In the revised version of the manuscript we added more information regarding the stepwise regression procedure. We also add a reference for the applied method.*

Page 5, Lines 10-25: - This paragraph belongs to either the introduction or the methods section. - Since you have given the citations of the data sets used already in the data set subsection there is not need to repeat them here.

*We removed parts of this paragraph, but we prefer to keep it in it's current location because we use it as an introduction to explain why we opt for the predictors we are using.*

Lines 11/12 vs. lines 14/15: You state that sea-ice cover and snow cover belong to the long-term memory components but then use OHC, OT100 and SST as long-term predictors. Wouldn't it be more straightforward to only refer to the oceanic components in Lines 11/12 and leave out snow cover and ice cover?

*Modified as suggested.*

Line 18: "if predictable" –> I don't understand this statement in this context. Why has the atmospheric circulation to be predictable if you aim for the prediction of September SIE based on spring atmospheric conditions?

*We deleted "if predictable" in the revised version of the paper.*

Line 21: What do you mean by "advective parameters"? - Lines 22-25: These two sentences should perhaps be re-formulating for clarity and to avoid repetition along the lines: "Atmospheric moisture content, e.g. clouds, water vapor content, has an impact on the net surface radiation

balance and hence also on the SSIE (Kapsch et al., 2013, 2014). As a measure for this impact we use the precipitable water content (PWC) as an additional predictor."

*Parts of this paragraph have been removed and some part have been modified as suggested.*

Page 5, Line 26 to Page 6, Line 4: - This paragraph belongs to either the introduction or the methods section.

*The paragraphs has been moved in the methods section.*

Line 1: "SSIE index" –> why index? I thought you are after the sea-ice extent? - Lines 2/3: "with different lags, depending on the variable" –> It seems you are using different time lags for different parameters in the correlation analysis? Please specify why. Please detail the time lags associated with which parameter. Line 3/4: "The optimal predictors are defined as the average values over the stable regions for each gridded parameter." –> unclear. Did you take a stability map between, say SSIE and an arbitrary parameter, e.g. SST, and average over the SST within the region defined as having a correlation at 90% significance? And then? Then you have an average SST value ... fine ... and next? Why is this the "optimal predictor"? Further, when doing this averaging, do you take into account the actual correlation value as well or do you only use the significance as a criterion to select over which SST values you are averaging?

*The text has been modified to properly capture the real meaning behind SSIE index. We are not using a gridded field, we are using just one time series, the September Sea Ice Extent, that's why we call it an index. Nevertheless, in the revised version of the manuscript we use the same definition throughout the manuscript, namely SSIE. Moreover, we have included also a table (Table 2) in which we clearly specify which time lags are involved for each variable and which kind of data (monthly and/or seasonal. This frames the analysis and helps to follow the method performed.*
*All the predictors used in the regression model are calculated based on the significance criterion. All the grid point in which the correlation coefficient between SSIE and large-scale data are significant at a significance level >90%, for more than 80% of the case, are used when defining an index (by averaging the gridded data over the black boxes in the stability maps).*
*In addition Table S1. showing the skill parameters (see supplementary file for definition) based on different statistical methods for the observed and predicted pan-Arctic sea ice extent in September with different time lags adds much information to understand the approach.*

Page 6, Lines 5-30: - It is not clear how you end up with the variables and time periods shown in Figures 1 through 3 and Table 1. It is not clear why in particular in some cases optimal predictors either are based on monthly values (of one or even two different months) or are based on seasonal averages; you have not introduces seasonal averages at all yet. This all seems quite arbitrary. - It it not clear what the criterion is to place black boxes in Figures 1 through 3 and what their meaning is. It is not clear in particular how the location of the black boxes go along with the notion that only in these stability maps only regions with > 90% significance are used. This all seems quite arbitrary.

*The method and approach is not all "arbitrary" as mentioned by the reviewer. Nevertheless, we think the reviewer is correct that we have to present the paragraph more structured and clearer concerning the analysis. In the revised version of the manuscript, we have added Figure S2 in the supplement, which shows the workflow of our methodology and how we end up with the "optimal model".*

-Figures 1 through 3: If the 90% significance level is as important as I think based on what you wrote, then I suggest to mark the 90% also by a change in color in the stability maps. Currently, 90% significance is right in the middle of (regular) red or blue. - In the captions of Figures 2 and 3 you need to state that these are the ADDITIONAL parameters (additional with respect to May, Figure 1) on which the prediction of the September sea-ice extent is based. - Lines 5/6: What is

meant by "...we have retained all the stable regions ... for all variables based on previous months' data."?

*The captions of Figure 2 and 3, as well as for ESS, have been modified following the reviewer's suggestion. Also the text has been modified in the revised version of the manuscript.*

Lines 11-22: The entire issue about AMO needs to be put into the methods section where you can introduce this as an important additional parameter. It should be introduced in the RESULTS section.

*The AMO paragraph has been moved in the discussion part.*

Figure 4: I suggest to change the y-axis title to "Sea-ice extent anomaly []"; currently it is "Sie ice extent []". This applies also to Figure 5. For Figure 5, I in addition note that the range of the y-axis differs between a) and b)&c); I suggest to make this consistent as there is no physical reason to have different ranges. Another comment, maybe a matter of taste, but I would not call your comparison of the last 7 years' skill of the prediction a "validation" - this is an evaluation. I would not call your 21-year period "calibration". Instruments like a pyranometer, pyrgeometer, anemometer, etc., are calibrated. You use that period to develop your method. Hence, I suggest to speak of a "development phase" or "training phase" and an "evaluation phase" It would be good, finally, to also show a graph of the difference between observation and prediction in relative terms; a difference of 300 000 km^2 with respect to the sea-ice extent of the entire Arctic is different in relative terms than a difference of 100 000 km^2 with respect to the ESS sea-ice extent. That way you can better quantify the skill of your method.

*Figure 4 and Figure 5 (now Figure 8) have been modified following the suggestions. Regarding the "calibration" and "validation" wording, we like to stress that this is the state of the art wording in forecasting. All the papers mentioned in the references, and not only, use the terms "calibration" and "validation" to train and evaluate their model. Thus, we choose to keep this wording also in the revised version of the manuscript. We also feel that by adding two new figures of the difference between observation and prediction in relative terms will not add much to the paper.*

Page 7, Lines 1-13: - Line 8: Please make clear that these are again additional parameters, i.e. on top of those for May (Fig. 1) and June (Fig. 2).

*Modified as suggested.*

Heading of section 3.2: I would reformulate the title of this subsection into: "Application of the methodology for regional SIE prediction" Also, my general view is that you should include all the supplementary figures and table of this subsection into the regular paper.

*We have modified the title of subsection 3.2 following the reviewer's suggestion. Moreover, we have now included all the former supplementary figures into the regular paper. In the actual supplementary document we have added new figures, for a better understanding of our stability regions determination.*

Lines 21-25: I don't think that Table 2 and these lines starting with "Moreover, when looking ..." are required to motivate your look at a specific region of the Arctic Ocean. A link / hint towards how you ended up at the optimal parameter combination would be an asset here as well for the correct understanding of your results (see also my comments with respect to Figures 1 through 3 and Table 1 further up).

*We have added a figure, which emphasizes the workflow of the selection of the optimal parameter combination (Figure S2 and additional text in the supplementary file).*

Page 8, Line 19 3/4: A discussion of the results is missing completely. See GC6

*The conclusion part has been modified substantially and now most of the discussion are integrated in the new part of the paper called Discussion and conclusions.*

Page 8, Line 21 to Page 9, Line 2: - Line 23: "Although" ... –> Why? - I suggest to move Line 24 - end of this paragraph to a later place in the conclusions. Generally, starting the conclusion with a short summary about what has been done in this study would put whatever information coming later in the conclusion into a better context.

*Modified as suggested.*

L28-30: I am not sure what the mentioning of the Parkinson et al (2006) paper and the excurse on modelling has to do with your work. After all, inconsistencies between model and observation with regard to sea-ice cover can have MANY reasons. The fact that OHC is one of these is neither new nor would I use that as one of the highlights to show why your study is valuable.

*The reference of Parkinson et al. (2006) is added as one example. We are aware the OHC is not a new highlight in term of predictability, but can be used to discuss potential issues.*

Page 9, Lines 3-24: This is part of the discussion - see GC6. - I suggest to split this paragraph into two in Line 16. the first one could talk about the teleconnections with, e.g. AMO and the impact the AMO has on which variable used in your method – if you find that relevant. Frankly speaking, if Yu et al. (2017) found that a strong link between these large scale pattern - why didn't you use correlations of between sea-ice cover and the strength of these patterns for your predictions? The second one could talk and refer to the regionally / locally driven feedbacks between ocean, sea-ice and atmosphere. You refer to SLP under these more regional forcings (Line 17) - but isn't particularly the SLP distribution the one which is directly linked to PDO, AMO, AO, NAO, etc.?

*As stated before, the conclusion part has been substantially modified and we added the discussion in this sub-section. We discuss the different issues raised here by the reviewer throughout the whole "Discussion and conclusion" sub-section.*

Line 4: What do you mean by "climate variables"?

*This wording has been removed from the revised version of the manuscript.*

Lines 4-6: - "(the stable regions over the western European coast)" –> I do not find these in your figures. - OHC: For most of the areas shown in Figure 1 a) the correlation with OHC of SON of the previous year (?) is stable with than 90% significance and hence does not enter your prediction according to your description of the method. Does this change for June & July as starting months of the prediction? - SST: Figure 1b) shows two areas of significant correlation with SST of MAM, a positive one in the Bering Sea and a negative one in the Barents Sea. You only used the one in the Bering Sea. Why?

*The text has been modified to properly capture what is found in the stability maps. As is stated in the description of the methodology we used all the regions which are above the 90% significance level. Moreover, we have added as supplementary file the original figures where we show all the stable regions used in the regressions model. What is shown is Figures 1 - 3 and Figures 5 – 8 are just the stable region that are used for the SSIE forecast after we applied the linear regression model.*

Line 10/11/12: Why "anomaly"? - Line 17 ++: I would about the "and so on" and I would try to be as explicit as possible here, referring to the respective Figure(s) for better understanding. It is crucial to discuss these influences in the light of which SLP distribution, associated VSURF and USURF distributions and PWC and TT patterns belong together physically and whether these are reflected in a consistent way in your stability maps. There is a lot to see in these stability maps and potentially also a lot to discuss albeit with the danger that one over-interprets correlations. In that respect you could improve the value of your paper and the discussion / conclusion in particular. A

discussion could and should include more information about your choice of regions used for the prediction (black boxes in Figures 1-3).

*To take into account this suggestion we have added a new section in the revised manuscript, namely Discussion (section 4), where we try to incorporate the reviewers comments/issues regarding a more detailed explanation of the physical mechanisms and some issues related to our methodology.*

Line 20-23: You mention PWC and air temperature in Lines 20/21 but then refer to "transport" in Line 23 - so how about the meridional transport of PWC?

*The text has been modified according to the suggestion.*

Page 9, Line 25 - Page 10, Line 5: - Line 28: 1996 and 2007 lie within your development or training phase while only 2012 lies outside it. Hence one could argue that the only true year for comparison is 2012 for which the agreement is not as good as for 1996, for example. You could include 2013 as well, for which the prediction is quite good. It is further interesting to see that for ESS a start of the prediction in July results in a substantially better agreement for the two years of extreme pan-Arctic minima 2007 and 2012 (compare Fig. 5c with Fig. 5 a). In contrast, for the entire Arctic it does not really make a difference whether you start in May (Fig. 4a) or July (Fig. 4c): the discrepancy between observations and prediction remains unchanged. But this of course belongs to the discussion, in which you need to critically assess your approach to here, in the conclusions, give explicit recommendations about which starting month and which parameters are suited best to achieve the best prediction. Lines 4-5: Yes, the concept can be used but how much better is it compared to other systems and concept. This you did not demonstrate and/or discuss. Therefore I would delete this last sentence. Another argument for deleting this sentence is that, e.g. shipping, might not be too much interested in the pan-Arctic sea-ice extent. Sea-ice area or, even better, the sea-ice distribution will be a better measure. As an example: The SSIE of the ESS is unchanged compared to winter but the sea-ice area reduced to 1/2 of the winter value if half of the ESS is ice covered by 75% sea ice and half of the ESS is ice covered by 25% sea ice. Hence the value of regional sea-ice extent prediction for shipping remains questionable.

*We have modified the revised version of the manuscript substantially trying to incorporate the suggestions of the reviewer. We have added a new section for discussion in which we discuss some of the issues as well as advantages of our methodology.*

Typos: - Please decide whether you use the American or British way of writing "skillful".
Page 2, L12: "flows" –> "floes"
*Modified as suggested.*

Page 3, Lines 20 & 24: "extracted". I suggest to write "obtained" or "downloaded".
*The respective line has been removed from the revised version of the manuscript.*

Page 4 Line 14: "level" –> "significance level" Line 24: "forecast m all" –> "m"? Line
*Modified as suggested.*

27: "each ... parameters" –> "each ... parameter"
*Modified as suggested.*

Page 5 Line 18: "substantial" –> "substantially"
*Modified as suggested.*

Page 7 Line 23: "EES" –> "ESS"
*Modified as suggested.*

Page 9 Line 11: "EES" –> "ESS" Line 12: "form" –> "from"
*Modified as suggested.*

[revised manuscript text omitted]

---

## Author Response (AR2)

I thank the authors for their tremendous effort to reply to my comments and to improve their manuscript. It is a reward if also the comments of a non-expert in statistical sea-ice extent forecast are taken serious.

I suggest that the paper is ready for publication pending the comments / technical corrections listed and correction of the typos.

**We thank the reviewer for the suggestions/comments/feedback that helped us improve our manuscript throughout the whole review process.**

Comments / Technical Corrections

On page 8, lines 11-13, you might want to add that increased cloudiness and humidity are responsible for enhanced melt via enhance longwave radiation. When talking about shortwave radiation budget, the conditions are not as straightforward as it seems (see Perovich, The Cryosphere, https://doi.org/10.5194/tc-12-2159-2018) because the efficiency of the impact of clouds on sea-ice melt strongly depends on timing and surface properties.

**The text has been modified following the reviewer's suggestion (page 8, lines 13-14)**.

Same page, lines 34-35: It is correct that the quoted storm possibly brought large amounts of heat and moisture but it might make sense to consider two other aspects which are presented here: Zhang J., R. Lindsay, A. Schweiger, and M. Steele (2013), The impact of an intense summer cyclone on 2012 Arctic sea ice retreat, Geophys. Res. Lett., 40, 720–726, doi:10.1002/grl.50190 --> the sea-ice cover was properly preconditioned already (see also Wang et al., doi:10.1002/2015JD023712) and the storm allowed a large amount of oceanic heat to be mixed up to the surface - which after all might have been more efficient to melt the sea ice than the heat in the atmosphere.

**Following the aforementioned suggestion we have modified the text accordingly and we also added new references (page 8, lines 36 – 40).**

Page 9, lines 6-9: It is very nice to see how your method captures this change in the overall atmospheric situation and provides an almost perfect prediction. May I nevertheless ask you to provide 1-2 references about the situation and/or the differences in the conditions in 2013 compared to 2012? Possible sources could be again the above-mentioned paper by Wang et al. as well as Liu and Key, Environmental Research Letters, doi:10.1088/1748-9326/9/4/044002.

**We have modified the text and provided the aforementioned references in the revised version of the manuscript (page 9, line 10 – 14).**

Typos:
On page 1:
IPPC --> IPCC

**Modified as suggested.**

On page 3:
ERSSTv4b --> ERSSTv5
RRSSTv5 --> ERSSTv5 (at least in your reply to the reviewers' comments you did mention that you used version 5 and that version 4 was a typo.)

**Modified as suggested.**